# Analyzing and Mitigating Inconsistency in Discrete Audio Tokens for Neural Codec Language Models

## Abstract

Building upon advancements in Large Language Models (LLMs), the field of audio processing has seen increased interest in training audio generation tasks with discrete audio token sequences. However, directly discretizing audio by neural audio codecs often results in sequences that fundamentally differ from text sequences. Unlike text, where text token sequences are deterministic, discrete audio tokens can exhibit significant variability based on contextual factors, while still producing perceptually identical audio segments. We refer to this phenomenon as **Discrete Representation Inconsistency (DRI)**. This inconsistency can lead to a single audio segment being represented by multiple divergent sequences, which creates confusion in neural codec language models and results in omissions and repetitions during speech generation. In this paper, we quantitatively analyze the DRI phenomenon within popular audio tokenizers such as EnCodec. Our approach effectively mitigates the DRI phenomenon of the neural audio codec. Furthermore, extensive experiments on the neural codec language model over LibriTTS and large-scale MLS dataset (44,000 hours) demonstrate the effectiveness and generality of our method. The demo of audio samples is available online [1].

## 1 Introduction

Recently, speech Large Language Models (LLMs) (Zhan et al., 2024; Anastassiou et al., 2024; Du et al., 2024b) have demonstrated significant strides in generating high-quality speech, largely due to the contributions of neural audio codecs in high-fidelity audio reconstruction (Zeghidour et al., 2021; Défossez et al., 2022; Yang et al., 2023). The neural codec language model (Wang et al., 2023; Yang et al., 2024; Zhang et al., 2024) employs the neural audio codec as the audio tokenizer to quantize continuous audio signals into discrete tokens, and it can generate discrete tokens autoregressively (Zhang et al., 2023a; Yang et al., 2024), and then detokenize them back to audio signals by the neural audio codec. Despite the advantages of autoregressive modeling can assist those works to achieve better zero-shot performance and naturalness, the synthesized speech frequently yields higher Word Error Rate (WER) due to the issue of instability in discrete token generation (Song et al., 2024; Xin et al., 2024; Du et al., 2024a).

The discrete sequence of text is context-independent. In contrast, acoustic discrete representations are encoded by integrating the contextual information. The advantage of this approach is that discrete audio tokens consider a larger receptive field of information, thus achieving a higher compression ratio of information. However, the drawback is that the representation itself becomes more fragile, sensitive, and easily affected by minor signal changes, leading to drastic drifts in the entire sequence as demonstrated in Figure 1.

The previous work (Yang et al., 2024) has noticed that audio segments containing the same sound events aren't encoded into completely consistent discrete acoustic tokens by the neural audio codec. In this paper, we call this phenomenon **Discrete Representation Inconsistency (DRI)**, and further dig into the problem with Vector Quantization (VQ) (Défossez et al., 2022) based acoustic tokens due to its popularity as an audio tokenizer and its high-quality reconstruction capabilities. We compare the consistency of the discrete sequences of audio segments with and without context on a large

---

[1] https://consistencyinneuralcodec.github.io

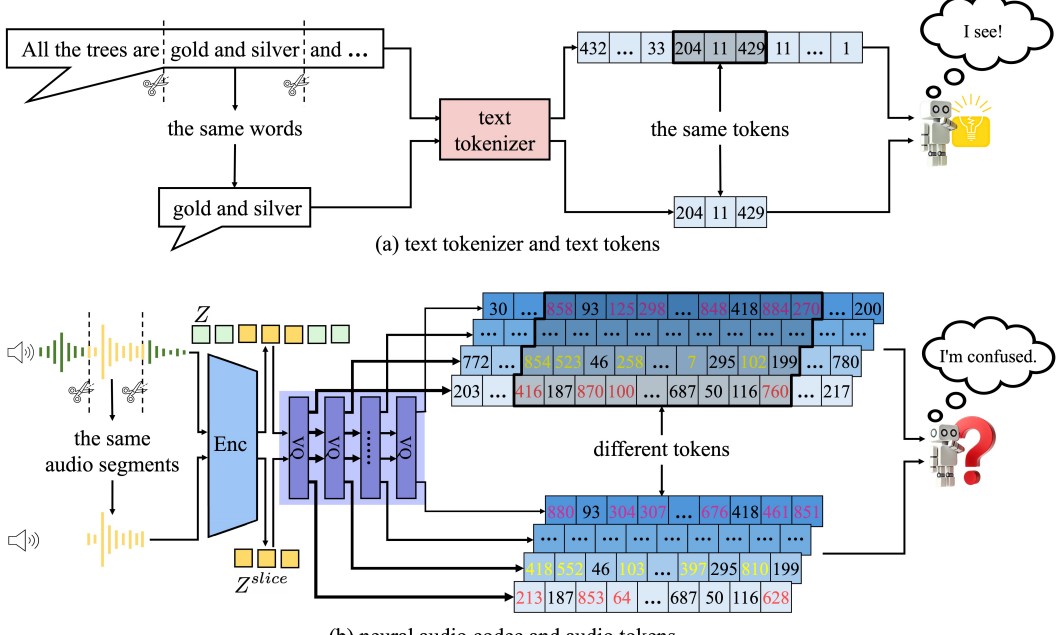

Figure 1: Discrete Representation Inconsistency (DRI) phenomenon. Subfigure (a) shows that text, whether it includes contextual information or not, can be encoded by the text tokenizer into identical text tokens. In contrast, Subfigure (b) illustrates that audio, with or without contextual information, is encoded by the audio tokenizer into different audio tokens. The DRI phenomenon within the audio tokenizer poses a many-to-one mapping problem, and the complexity of this many-to-one mapping raises the uncertainty for neural codec language models in predicting the next token.

amount of audio. Our quantitative analyses reveal that the existing audio tokenizers suffer from low consistency. In particular, we find that for Residual Vector Quantization (RVQ) (Défossez et al., 2022) approaches, consistency declines significantly with deeper layers of codebooks.

Although audio with or without contextual audio is encoded into different discrete audio token sequences, both sequences can be used to reconstruct the original audio information, which leads to a many-to-one mapping problem that becomes more complex as the sequence length increases. This complexity results in increased uncertainty for neural codec language models in predicting the next token. One direct approach to address this issue is to prevent the audio tokenizer from considering contextual information during sequence encoding, such as by setting the convolutional layer's kernel size to 1 in the encoder. While this method allows for independent discretization of each audio frame, it also significantly reduces encoding efficiency and degrades the quality of the reconstructed audio. Therefore, this study aims to maintain the original receptive field while enabling the model to address the trade-offs between audio reconstruction quality and resolving the many-to-one problem. To achieve this objective, we introduce the **slice-consistency** method, wherein a segment of audio is randomly sliced, and the encoded representation from this sliced segment is required to closely approximate the corresponding representation obtained from the entire audio. In addition, in order to further alleviate the issue of many-to-one mapping, we propose the **perturbation-consistency** method, whereby the representation of an audio and its representation after applying slight spectral perturbation should closely align. Compared to EnCodec (Défossez et al., 2022), our method has shown an average consistency improvement of 21.47%, 29.17%, and 36.29% in the first layer, the first 3 layers, and the first 8 layers, respectively. Extensive experiments on the neural codec language model (e.g., VALL-E (Wang et al., 2023)) on LibriTTS (Zen et al., 2019) dataset (960 hours) and large-scale MLS (Pratap et al., 2020) dataset (44,000 hours) confirm that improving consistency results in better performance. Our contributions are summarized as below:

- We shed light on the Discrete Representation Inconsistency (DRI) phenomenon and conduct quantitative analyses for various neural audio codecs. We find that the existing audio tokenizers suffer from low consistency.

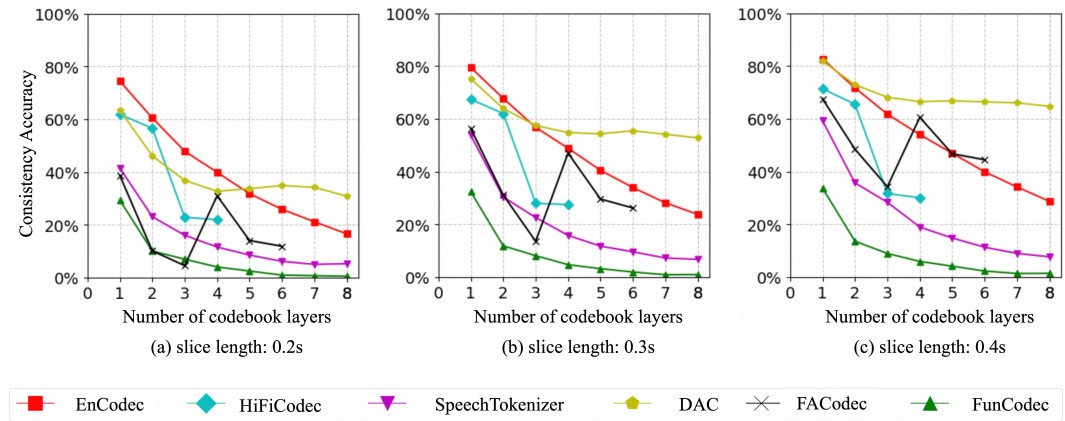

Figure 2: Results of consistency accuracy for popular neural audio codecs under different layers and slice lengths. Subfigures (a), (b) and (c) show slice lengths across 0.2s, 0.3s and 0.4s, respectively, and all of them exhibit similar conclusions that consistency accuracy declines significantly in the deeper layers of codebooks, indicating that the DRI phenomenon becomes more pronounced with layers in neural audio codecs increasing.

- Inspired by our analyses, we propose two methods, the slice-consistency method and the perturbation-consistency method, to enhance the consistency of the neural audio codec from two particular perspectives and mitigate the many-to-one problem.
- Experiments show that our method achieves an average consistency improvement of 21.47%, 29.17%, and 36.29% in the first layer, the first 3 layers, and the first 8 layers, respectively. Additionally, we conduct extensive experiments on the neural codec language model, VALL-E, on the LibriTTS dataset (960 hours) and further expand the training dataset to the large-scale MLS dataset (44,000 hours), resulting in 3.72% WER reduction and 5.68% speaker similarity improvement. These findings confirm that enhancing consistency leads to improved performance.

## 2 ANALYSIS ON CONSISTENCY OF NEURAL AUDIO CODECS

In this section, we extract discrete audio tokens from audio segments with and without context using popular neural audio codecs (Défossez et al., 2022; Yang et al., 2023; Zhang et al., 2023b; Du et al., 2024c; Kumar et al., 2024; Ju et al., 2024) to analyze the DRI phenomenon. First, we introduce the overall experiment design. Then we propose using **consistency accuracy** as an evaluation metric to conduct quantitative analyses. Finally, we analyze the results and discuss the potential implications of the DRI phenomenon.

### 2.1 EXPERIMENTAL DESIGN ON DRI PHENOMENON

Recent advancements on neural audio codecs have adopted an encoder-decoder architecture combined with the RVQ module to effectively compress continuous audio signals into discrete audio tokens (Défossez et al., 2022; Yang et al., 2023; Zhang et al., 2023b; Du et al., 2024c; Kumar et al., 2024; Ju et al., 2024), which is typically composed of 3 components: (1) An encoder, composed of convolutional layers to capture contextual information, maps the audio signal into a latent representation $Z$. (2) An RVQ module contains $N$ quantization layers to quantize the latent representation $Z$ into the discrete audio tokens at each time step. (3) A decoder reconstructs the quantized latent representation back to the audio signal.

To analyze the DRI phenomenon, we use popular neural audio codecs (Défossez et al., 2022; Yang et al., 2023; Zhang et al., 2023b; Du et al., 2024c; Kumar et al., 2024; Ju et al., 2024) as audio tokenizers to quantize both the entire audio and an audio segment within that audio, and then compare the results of their corresponding discrete audio tokens. Obviously, these two audio segments are exactly identical with the only difference being whether there is context, and we expect that both

discrete audio tokens should be identical after quantization. But the encoders in current neural audio codecs introduce the contextual information that gives rise to the DRI phenomenon, leading to both discrete audio token sequences showing significant differences.

## 2.2 Consistency Accuracy

To quantitatively analyze the degree of the DRI phenomenon in neural audio codecs, we propose using **consistency accuracy** as an evaluation metric:

$$Acc_{\text{consistency}} = \frac{1}{T}\frac{1}{N}\sum_{t=1}^{T}\sum_{i=1}^{N}\mathbb{I}(\text{RVQ}(Z^{\text{slice}})[t,i] = \text{RVQ}(Z)[t,i]), \tag{1}$$

where $Z$ is the latent representation of the original audio after encoding by the encoder, and $N$ represents the number of codebooks in the RVQ module. We randomly extract an audio segment of length $T$ from the original audio, and encode it by the encoder to obtain $Z^{\text{slice}}$.

## 2.3 Results And Analysis

**Audio tokenizer vs. text tokenizer**. As shown in Figure 1 (a), regardless of whether the context is included, the same text is tokenized into the same text tokens, indicating that the text tokenizer is context-independent. In contrast, Figure 1 (b) demonstrates that using a neural audio codec as the audio tokenizer produces different discrete audio token sequences for identical audio segments. Although it is difficult for human auditory perception to distinguish the reconstructed audio from both sequences, the many-to-one mapping caused by the DRI phenomenon still increases the difficulty for model training, leading to a decline in speech reconstruction and generation performance.

**The results of consistency accuracy**. To quantitatively analyze the DRI phenomenon, we calculate the consistency accuracy for popular neural audio codecs under different layers and slice lengths. The results are shown in Figure 2 and the low consistency accuracy reveals that the DRI phenomenon is present in the current neural audio codecs (Défossez et al., 2022; Yang et al., 2023; Zhang et al., 2023b; Du et al., 2024c; Kumar et al., 2024; Ju et al., 2024). Furthermore, we find that with deeper layers of codebooks, neural audio codecs demonstrate lower consistency. This may be attributed to the fact that audio tokens in shallow layers exhibit a high alignment with context-independent semantic information, resulting in better consistency. In contrast, deeper layers focus on more fragile and sensitive acoustic information that can easily change due to minor perturbations, leading to a decrease in consistency accuracy (Zhang et al., 2023b).

**The potential implications of the DRI phenomenon**. There are many minor perturbations that can cause the DRI phenomenon, such as contextual information and phase perturbation (Lee et al., 2023) that do not alter the auditory perception of the reconstructed audio but can lead to changes in the discrete audio token sequences, which can greatly confuse models. Especially when neural codec language models need to predict different audio tokens due to the DRI phenomenon, this confusion can cause the predicted probability distributions of the next token to converge towards uniformity, resulting in inaccurate predictions and negatively impacting overall performance. Therefore, it is necessary to ease the many-to-one mapping problem to improve the consistency of neural audio codecs, which in turn enhances the performance of downstream speech generation.

## 3 Method

According to the analysis in Section 2, we can draw a conclusion that an ideal neural audio codec should balance the trade-offs between high audio reconstruction quality and addressing the many-to-one problem. To achieve this objective, we introduce two consistency constraint methods: the **slice-consistency** method and **perturbation-consistency** method, which enhance the consistency of the neural audio codec from two particular perspectives. Since these methods can be integrated into any neural audio codec, we demonstrate their application using a neural audio codec based on RVQ which utilizes an encoder to transform the audio signal into the latent representation $Z$ and reconstructs the waveform from the quantized latent representation.

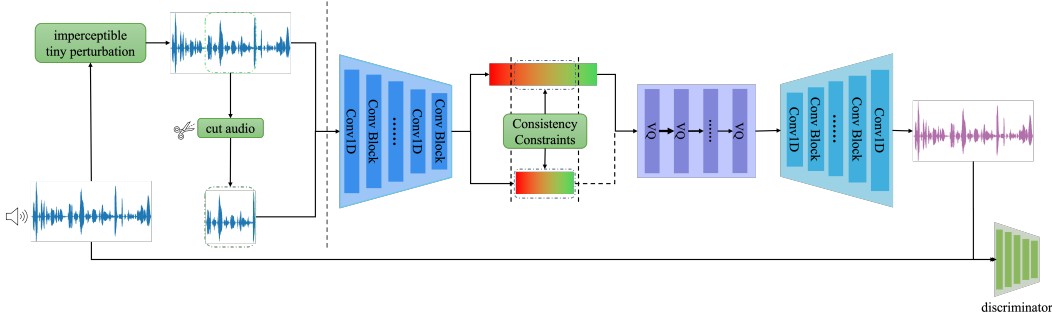

Figure 3: The overview of the proposed consistency constraint method. For the slice-consistency method, a segment of audio is randomly sliced, and its encoded representation must closely match the representation derived from the entire audio. For the perturbation-consistency method, the representation of an audio and its representation after slight spectral perturbation should be closely aligned.

## 3.1 CONSISTENCY CONSTRAINT METHODS

**Slice-consistency** requests that audio segments with and without context should be encoded into consistent latent representations. To achieve this object, as shown in Figure 3, we slice a segment of audio from the original audio, and then encode it using the encoder in the neural audio codec to obtain the latent representation $Z^{\text{slice}}$. Compared with the latent representation $Z$ from the entire audio, $Z^{\text{slice}}$ is not influenced by contextual information. To reduce the influence of context on the latent representation $Z$, we use Mean Squared Error (MSE) as a constraint to enhance the consistency between $Z^{\text{slice}}$ and the corresponding latent representation in $Z$:

$$\mathcal{L}_{\text{slice}} = \frac{1}{T} \sum_{t=1}^{T} \text{MSE}(Z^{\text{slice}}[t], Z[t]). \tag{2}$$

As analyzed in Appendix 8.1 about the receptive field, the convolutional layers in the encoder of neural audio codecs introduce contextual information, leading to identical audio segments being tokenized into different discrete audio token sequences. It is clear that reducing the kernel size of the convolutional layers in the encoder can enhance consistency, but this can also result in a decline in both reconstruction efficiency and quality. Therefore, applying the slice-consistency method is necessary to maintain the original receptive field while enabling models to balance the trade-offs between audio reconstruction quality and alleviating the DRI phenomenon.

**Perturbation-Consistency** refers to the latent representations of audio, which should remain consistent before and after being applied imperceptible perturbations to human ears. Specifically, as shown in Figure 3, we slightly adjust the phase of the the original audio without significantly altering the waveform structure, and encode it using the encoder in the neural audio codec to obtain the latent representation $Z^{\text{perception}}$. Since human ears have a limited ability to directly perceive phase changes, we hope that the robustness of the model can also eliminate inconsistency caused by such slight perturbations. Therefore, we utilize MSE to maintain consistency of both latent representations with and without phase perturbation (Lee et al., 2023):

$$\mathcal{L}_{\text{perception}} = \text{MSE}(Z^{\text{perception}}, Z). \tag{3}$$

It is evident that the perturbation-consistency method differs from audio-based data augmentation methods. Data augmentation methods such as SpecAugment (Park et al., 2019) and environment noise (Snyder et al., 2015) significantly alter the original audio to create new audio. The newly generated audio has a considerable difference in perception compared to the original audio, which aims to expand the training data and increase the robustness of models. In contrast, the perception-consistency method requires that changes to audio should be imperceptible to human ears to avoid severe perturbations that disrupt the audio reconstruction quality. Since the phase is difficult to be

perceived by human, we apply phase perturbation (Lee et al., 2023) as a slight perturbation method, which can enhance the perturbation-consistency without expanding the training data.

## 3.2 IMPLEMENTATION DETAILS

In order to satisfy both methods and enhance training efficiency, we align the latent representation $Z^{perception}$ obtained by the slice-consistency method and the latent representation $Z^{perception}$ obtained by the perturbation-consistency method:

$$\mathcal{L}_{\text{consistency}} = \frac{1}{T} \sum_{t=1}^{T} \text{MSE}(Z^{\text{slice}}[t], Z^{\text{perception}}[t]). \tag{4}$$

By introducing consistency constraint $\mathcal{L}_{\text{consistency}}$, our method can be applied to any neural audio codec and we build our method on RVQ-GAN framework (Kumar et al., 2024) that also includes reconstruction loss $\mathcal{L}_{\text{rec}}$, adversarial loss $\mathcal{L}_{\text{adv}}$, feature matching loss $\mathcal{L}_{\text{fm}}$, and commit loss $\mathcal{L}_{\text{rvq}}$:

$$\mathcal{L} = \mathcal{L}_{\text{rec}} + \lambda_{\text{adv}}\mathcal{L}_{\text{adv}} + \lambda_{\text{fm}}\mathcal{L}_{\text{fm}} + \lambda_{\text{rvq}}\mathcal{L}_{\text{rvq}} + \lambda_{\text{con}}\mathcal{L}_{\text{consistency}}. \tag{5}$$

# 4 EXPERIMENT SETTING

## 4.1 EXPERIMENTAL CONFIGURATION

**Datasets**. During the training process of the neural audio codec and the neural codec language model, we utilize LibriTTS (Zen et al., 2019) dataset, which comprises 960 hours of transcribed speech data. The test set of LibriTTS (Zen et al., 2019) is used as test data, which is randomly selected 2,300 audio samples to verify the consistency of neural audio codecs and 350 audio samples to validate speech generation performance. To verify the effectiveness of data scaling on our proposed method, we expand the training dataset to 44,000 hours from large-scale MLS (Pratap et al., 2020) dataset for both speech reconstruction and speech generation tasks.

**Training settings**. To validate effectiveness of consistency constraint in speech reconstruction, we apply it on the RVQ-based neural audio codec (denoted as **Ours**) that uses the Adam optimizer (Diederik, 2014), with an initial learning rate of 3e-4 and beta parameters set to (0.5, 0.9), to train for 350,000 iterations. All audio samples are truncated to a fixed length of 1.28 seconds and resampled to 16 kHz with the batch size of 384. In the loss function 5, the weights are set as $\lambda_{adv} = 0.11$, $\lambda_{fm} = 11.11$, $\lambda_{rvq} = 1.0$, and $\lambda_{con} = 10.0$ when consistency constraint is applied.

To demonstrate the effectiveness of our method for the downstream speech generation task, we take the neural audio codec based on our method as the audio tokenizer for the neural codec language model, VALL-E (Wang et al., 2023) that generates audio by predicting the first layer of audio tokens in an autoregressive manner, and then predicting the remaining audio tokens in a non-autoregressive manner. The reproduced VALL-E (Wang et al., 2023) is trained for 1,300,000 steps with the batch size of 56 and optimized by the Adam optimizer (Diederik, 2014), with parameters $\beta$ = (0.9, 0.95).

**Baseline models**. For speech reconstruction, we use the official open-source checkpoints from EnCodec (Défossez et al., 2022), HiFiCodec (Yang et al., 2023), SpeechTokenizer (Zhang et al., 2023b), DAC (Kumar et al., 2024), and FunCodec (Du et al., 2024c) as baseline models. To ensure fair comparison, we set the bandwidth of different neural audio codecs closely to 4.0 kbps or 8.0 kbps. For speech generation, we employ the SOTA neural codec language models as baselines, including SpeechGPT (Zhang et al., 2023a), SpeechTokenizer-based USLM (Zhang et al., 2023b), AnyGPT (Zhan et al., 2024), VoiceCraft (Peng et al., 2024) and XTTS v2 (Casanova et al., 2024).

## 4.2 EVALUATION METRICS

### 4.2.1 EVALUATION OF SPEECH RECONSTRUCTION

We adopt consistency accuracy across all layers of neural audio codecs to measure the DRI phenomenon. Given that the codewords in the first few layers of the neural audio codec store the most information, their consistency significantly affects the performance of downstream neural codec language models. Therefore, we especially present the first 3 layers' consistency accuracy. Since the

Table 1: The speech reconstruction results on LibriTTS test set. **Bold** means the best result, and underline means the second-best result. **Ours** denotes the neural audio codec with consistency constraint. The subscripts of the neural audio codecs denote the training data scale.

| Neural Audio Codec | Bandwidth | Sampling Rate | Number of Codebooks | Consistency↑ | First 3 Layers' Consistency↑ | ViSQOL↑ | PESQ↑ |
|---|---|---|---|---|---|---|---|
| EnCodec$_{2690h}$ | 4.5 kbps | 24kHz | 6 | 47.43% | 61.49% | 4.25 | 2.41 |
| | 6.0 kbps | | 8 | 40.46% | 61.49% | 4.35 | 2.73 |
| | 8.25 kbps | | 11 | 32.77% | 61.49% | 4.44 | 3.02 |
| HiFiCodec$_{1122h}$ | 3.0 kbps | 24kHz | 4 | 40.77% | 46.92% | 4.32 | 2.76 |
| SpeechTokenizer$_{960h}$ | 4.0 kbps | 16kHz | 8 | 14.70% | 26.91% | 4.36 | 2.62 |
| DAC$_{2740h}$ | 4.0 kbps | 16kHz | 8 | 39.14% | 48.43% | 4.44 | 2.68 |
| FunCodec$_{960h}$ | 4.0 kbps | 16kHz | 8 | 6.86% | 16.39% | 4.47 | 3.26 |
| | 8.0 kbps | | 16 | 3.58% | 15.49% | 4.57 | **3.62** |
| Ours$_{960h}$ | 4.0 kbps | 16kHz | 8 | **71.03%** | 88.82% | 4.45 | 3.25 |
| | 8.0 kbps | | 16 | 56.32% | **90.66%** | **4.64** | 3.59 |

conclusions obtained from different lengths are generally consistent, we set $T$ in the consistency accuracy to 0.2. In addition, we also use ViSQOL (Chinen et al., 2020) and PESQ (Rix et al., 2001) to measure the quality of the reconstructed speech (Défossez et al., 2022; Zeghidour et al., 2021), with higher scores indicating the better speech quality.

### 4.2.2 EVALUATION OF SPEECH GENERATION

**Objective evaluation**. We use Whisper (Radford et al., 2023) model to transcribe the generated speech and calculate the WER. To evaluate speaker similarity, we firstly use 3D-speaker (Chen et al., 2024) toolkit to extract speaker embeddings from the generated speech and reference speech, and then compute the cosine similarity between the normalized embeddings. We also employ UT-MOS (Saeki et al., 2022) as an automatic Mean Opinion Score (MOS) prediction system to assess the naturalness of the speech.

**Subjective evaluation**. We randomly select 50 audio samples from the LibriTTS (Zen et al., 2019) test set to conduct MOS (Chu & Peng, 2006) and Similarity Mean Opinion Score (SMOS) (Chu & Peng, 2006) test. MOS assesses the naturalness of the generated speech, while SMOS measures the similarity between the generated speech and the original speaker's voice. Both MOS and SMOS range from 1 to 5, with higher values indicating better speech quality and greater speaker similarity.

## 5 RESULT

In this section, we evaluate the effectiveness of the slice-consistency method and the perturbation-consistency method. Firstly, we compare the speech reconstruction results of the neural audio codec wiht consistency constraint and baseline models. Then, we demonstrate that introducing our method to speech generation model like VALL-E (Wang et al., 2023) can effectively enhance speech generation performance on both small-scale and large-scale data. Finally, we conduct ablation studies to illustrate the effects of the slice-consistency method and the perturbation-consistency method, respectively.

### 5.1 SPEECH RECONSTRUCTION RESULTS

We evaluate the effectiveness of our method from the perspectives of consistency and reconstructed speech quality. First, we compare the consistency accuracy between the neural audio codec with consistency constraint and baseline models. The results presented in Table 1 demonstrate that the neural audio codec based on our method can reconstruct speech with superior consistency accuracy compared to baseline models, achieving 71.03% across all layers at the bandwidth setting of 4.0 kbps and 90.66% across the first 3 layers at the bandwidth setting of 8.0 kbps. In contrast, the baseline models suffer from low consistency accuracy, indicating that the same audio segments are encoded into different discrete audio token sequences.

Table 2: The speech generation results on LibriTTS test set. **Bold** means the best result, and underline means the second-best result. **Ours** and **Ours w/o consistency constraint** denote the same neural audio codecs with and without consistency constraint. The subscripts of the neural codec language models (e.g., $330M, 44Kh$) denote the model size and data scale.

| Neural Audio Codec | Bandwidth | Neural Codec Language Model | WER↓ | Objective SIM↑ | UTMOS↑ | Subjective MOS↑ | SMOS↑ |
|---|---|---|---|---|---|---|---|
| Ground Truth | / | / | 1.37 | / | 4.15 | 4.43 | 4.23 |
| mHuBERT | 0.5 kbps | SpeechGPT$_{72Kh}$ | 13.39 | 11.87% | 4.10 | 3.08 | 1.63 |
| EnCodec | 2.2 kbps | VoiceCraft$_{330M,9Kh}$ | 2.57 | 71.05% | 3.55 | 3.58 | 3.47 |
| | | VoiceCraft$_{830M,9Kh}$ | 2.80 | 78.26% | 3.76 | 3.72 | 3.43 |
| Mel VQ-VAE | / | XTTS_v2$_{27Kh}$ | 1.64 | 83.96% | 3.92 | 3.58 | 3.85 |
| SpeechTokenizer | 4.0 kbps | USLM$_{960h}$ | 6.86 | 43.36% | 3.05 | 3.07 | 2.90 |
| | | AnyGPT$_{57Kh}$ | 18.93 | 34.60% | 3.15 | 2.77 | 2.63 |
| Ours w/o consistency constraint | 4.0 kbps | VALL-E$_{960h}$ | 4.73 | 76.95% | 4.10 | 3.73 | 3.50 |
| | | VALL-E$_{44Kh}$ | 5.09 | 78.46% | 4.14 | 3.92 | 3.40 |
| Ours | 4.0 kbps | VALL-E$_{960h}$ | 1.84 | 83.71% | **4.31** | 3.97 | 3.73 |
| | | VALL-E$_{44Kh}$ | **1.37** | **84.14%** | 4.30 | **4.02** | **3.95** |

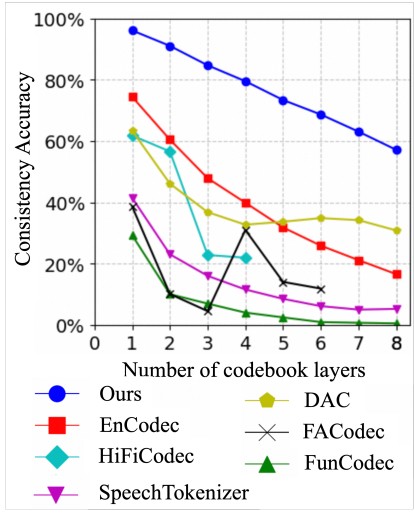

Figure 4: Consistency accuracy of each layer in neural audio codecs. **Ours** denotes the neural audio codec with consistency constraint.

As shown in Figure 4, we can observe that consistency accuracy declines significantly in the deeper layers of codebooks, particularly in baseline models. This observation may stem from the fact that the semantic information in the shallow layers of codebooks is relevant to text that is more context-independent, resulting in higher consistency accuracy. In contrast, the acoustic information in the deeper layers is more fragile and sensitive, making it more susceptible to contextual influences, which may pose challenges for downstream neural codec language models when predicting audio tokens from these deeper layers (Zhang et al., 2023b). More details about the accuracy of each layer can be found in Appendix 8.4. Then we evaluate ViSQOL (Chinen et al., 2020) and PESQ (Rix et al., 2001) to evaluate the reconstructed speech quality. The results in Table 1 show that the ViSQOL (Chinen et al., 2020) of the neural audio codec based on our method surpasses all baseline models, achieving the score of 4.64. Additionally, its PESQ (Rix et al., 2001) is also comparable to that of the baseline models, with only 0.03 lower than the best result. This suggests that our method can be confidently applied to neural audio codecs without negatively impacting reconstruction performance.

## 5.2 SPEECH GENERATION RESULTS

In this section, we utilize the neural audio codec both with and without the consistency constraint to replicate VALL-E (Wang et al., 2023). From both subjective and objective perspectives, we evaluate the generated speech to demonstrate the effectiveness of our method for the downstream speech generation task. Furthermore, we increase the training data from 960 hours of LibriTTS (Zen et al., 2019) dataset to 44,000 hours of large-scale MLS (Pratap et al., 2020) dataset to verify that our method is equally effective on larger scale datasets.

**Objective Evaluation**. According to Table 2, we have the following observations: (1) VALL-E (Wang et al., 2023), which is based on our method and trained by large-scale MLS (Pratap et al., 2020) dataset, outperforms all other baseline models on WER, SIM and UTMOS, indicating that our method can help speech generation models synthesize speech with better intelligibility, sim-

Table 3: Ablation study on the slice-consistency method and perturbation-consistency method. In the **Slice** column, the percentage (e.g., 20%) represents the proportion of the sliced audio segments to the entire audio. In the **Perturbation** column, **phase perturb** means whether to use perturbation-consistency method.

| | Neural Audio Codec | | | Neural Codec Language Model | | |
|---|---|---|---|---|---|---|
| Slice | Perturbation | Consistency↑ | First 3 Layers' Consistency↑ | Objective | | |
| | | | | WER↓ | SIM↑ | UTMOS↑ |
| 20% | phase perturb | 76.75% | 90.66% | 1.84 | 83.71% | 4.31 |
| / | phase perturb | 7.03% | 16.20% | 2.24 | 77.09% | 4.15 |
| 20% | / | 75.91% | 90.85% | 2.36 | 81.84% | 4.14 |
| / | / | 6.94% | 15.49% | 4.73 | 76.95% | 4.10 |
| 40% | phase perturb | 64.74% | 85.44% | 1.90 | 82.81% | 4.27 |
| 60% | phase perturb | 31.79% | 60.95% | 3.02 | 82.41% | 4.25 |

ilarity and naturalness. (2) Compared to the VALL-E model without the consistency constraint, our method can help VALL-E achieve significant improvement in intelligibility and similarity, with 3.72% WER reduction and 5.68% SIM improvement. This indicates that improving the consistency of the neural audio codec can reduce the complexity of predicting discrete audio tokens and result in better performance. (3) The results show that VALL-E (Wang et al., 2023), which is based on our method and trained by 44,000 hours, shows superior speech generation results than that trained on 960 hours, achieving the average reduction of 0.47 in WER and improvement of 0.43% in SIM, illustrating the scalability of our method across different dataset scales.

**Subjective Evaluation**. In subjective evaluation, we conduct MOS and SMOS tests to assess speech quality and speaker similarity for all of the neural codec language models, as shown in Table 2. The results of MOS and SMOS show similar outcomes to objective evaluations, indicating that VALL-E (Wang et al., 2023) based on our method achieves higher speech quality and speaker similarity.

## 5.3 ABLATION STUDY

In this section, we conduct ablation experiments to verify the effects of the slice-consistency method and the perturbation-consistency method, respectively. As shown in Table 3, we use the case of slicing the audio at 20% and applying perturbation-consistency method as a reference, which achieves the best results in both speech reconstruction and speech generation. Then we remove the design of slice-consistency method or perturbation-consistency method. The drop in all evaluation metrics demonstrates that both slice-consistency method and perturbation-consistency method are beneficial for speech reconstruction and generation. Finally, we conduct ablation studies on the proportion of slicing audio segments, and the results show that the slice percentage of 20% outperforms the model with the slice percentages of 40% and 60%. This suggests that shorter audio segments containing less contextual information can effectively alleviate the contextual dependence of original audio representation during the alignment process, thereby enhancing its consistency and ultimately leading to better performance in the downstream speech generation model. We also observed that although the consistency improvement brought by phase perturbation is small, it brings significant improvement to the downstream speech generation performance. We think that applying phase perturbation alone may help decouple information within the audio by altering the structure of acoustic information, thereby preventing the model from overfitting to unimportant features. This suggests that consistency is not the sole determining factor for speech generation performance.

Considering the better consistency in shallow layers of codebooks, we further analyze VALL-E (Wang et al., 2023) based on our method with fewer codebooks in Appendix 8.5. The experimental results demonstrate that our method is effective across different numbers of codebooks, indicating the generalizability of the proposed method.

## 6 RELATED WORK

**Discrete speech representations**. Discrete speech representations can be categorized into semantic and acoustic tokens. Discrete semantic tokens are extracted from self-supervised speech models like HuBERT (Hsu et al., 2021) and WavLM (Chen et al., 2022), or Automatic Speech Recognition (ASR) models like SenseVoice (SpeechTeam, 2024). K-means or VQ models serve as information bottlenecks, filtering out paralinguistic information while retaining semantic information. In contrast, discrete acoustic tokens are encoded by neural audio codecs, preserving complete acoustic information and aiming to reconstruct high-fidelity audio. SoundStream (Zeghidour et al., 2021) and EnCodec (Défossez et al., 2022) adopt RVQ framework to encode speech into multi-level discrete acoustic tokens. SingleCodec (Li et al., 2024) and Disen-TF-Codec (Jiang et al., 2023) utilize a reference encoder to capture global time-invariant information, thereby reducing the number of codebooks. SpeechTokenizer (Zhang et al., 2023b) and FACodec (Ju et al., 2024) decouple speech into different attributes, making discrete tokens more suitable for downstream speech modeling tasks. However, the synthesized speech from the neural codec language model, which relies on discrete audio tokens, often leads to a higher WER (Song et al., 2024; Xin et al., 2024; Du et al., 2024a). This is because these discrete audio tokens are fragile and sensitive, easily affected by minor changes in the audio signal. Inspired by the context-independent text tokens, we propose enhancing the consistency of audio token sequences to address the many-to-one mapping problem and improve the stability of discrete audio tokens.

**Audio tokenizers and neural codec language models**. After tokenizing continuous audio signals into discrete tokens by a neural audio codec, a neural codec language model can be trained on these discrete audio tokens. VALL-E (Wang et al., 2023) and SpearTTS (Kharitonov et al., 2023) employ EnCodec (Défossez et al., 2022) and SoundStream (Zeghidour et al., 2021) as audio tokenizers to extract discrete acoustic tokens, aiming to retain all acoustic information. SpearTTS (Kharitonov et al., 2023) and SoundStorm (Borsos et al., 2023) adopt a coarse-to-fine approach to generate discrete acoustic tokens. VoiceCraft (Peng et al., 2024) rearrange audio tokens through an autoregressive way to perform speech generation and editing tasks. LLM-Codec (Yang et al., 2024) represents audio tokens with words or subwords from the vocabulary of LLMs, aligning audio modality with text modality. Although LLM-Codec (Yang et al., 2024) has noticed that even when audio segments contain the same sound events, the discrete tokens generated by the audio tokenizer may still exhibit inconsistency. Therefore, to address this DRI phenomenon, we propose the slice-consistency method and perturbation-consistency method to enhance the consistency within neural audio codecs, thereby improving the performance of downstream speech generation.

## 7 CONCLUSION

We conduct a detailed analysis on the consistency of the discrete audio token sequences, and shed light on the Discrete Representation Inconsistency (DRI) phenomenon within the existing neural audio codecs. To mitigate the DRI phenomenon, we propose two consistency enhancement methods: (1) The slice-consistency method requires that the representation from a randomly sliced audio segment should match the corresponding representation from the entire audio. (2) The perturbation-consistency method aims to align the representation obtained from the audio after applying slight spectral perturbations with that from the original audio. Experimental results indicate that our proposed methods can successfully increase the consistency of discrete audio token sequences, thereby enabling the neural codec language model based on these audio tokens to outperform the SOTA speech generation model and show better performance by scaling up data. For future work, we plan to try more consistency enhancement methods and apply our method to various modalities to assess its generalizability.

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

# 8 APPENDIX

## 8.1 ANALYSIS OF INCONSISTENCY CAUSED BY RECEPTIVE FIELD SIZES

Table 4: The parameters of the convolutional layers and the receptive field size in the neural audio codec's encoder.

| Layer ID | Kernel Size | Stride | Dilation | Strides of Previous Layers | Receptive Field Size |
|---|---|---|---|---|---|
| 1 | 7 | 1 | 1 | 0 | 7 |
| 2 | 3 | 1 | 1 | 1 | $7 + (3-1) \times 1 = 9$ |
| 3 | 1 | 1 | 1 | 1 | $9 + (1-1) \times 1 = 9$ |
| 4 | 1 | 1 | 1 | 1 | $9 + (1-1) \times 1 = 9$ |
| 5 | 4 | 2 | 1 | 1 | $9 + (4-1) \times 1 = 12$ |
| 6 | 3 | 1 | 1 | 2 | $12 + (3-1) \times 2 = 16$ |
| 7 | 1 | 1 | 1 | 2 | $16 + (1-1) \times 2 = 16$ |
| 8 | 1 | 1 | 1 | 2 | $16 + (1-1) \times 2 = 16$ |
| 9 | 8 | 4 | 1 | 2 | $16 + (8-1) \times 2 = 30$ |
| 10 | 3 | 1 | 1 | 8 | $30 + (3-1) \times 8 = 46$ |
| 11 | 1 | 1 | 1 | 8 | $46 + (1-1) \times 8 = 46$ |
| 12 | 1 | 1 | 1 | 8 | $46 + (1-1) \times 8 = 46$ |
| 13 | 10 | 5 | 1 | 8 | $46 + (10-1) \times 8 = 118$ |
| 14 | 3 | 1 | 1 | 40 | $118 + (3-1) \times 40 = 198$ |
| 15 | 1 | 1 | 1 | 40 | $198 + (1-1) \times 40 = 198$ |
| 16 | 1 | 1 | 1 | 40 | $198 + (1-1) \times 40 = 198$ |
| 17 | 16 | 8 | 1 | 40 | $198 + (16-1) \times 40 = 798$ |
| 18 | 7 | 1 | 1 | 320 | $798 + (7-1) \times 320 = 2718$ |

The size of the receptive field is related to the number of convolutional layers and pooling layers:

$$\text{RF}_i = \text{RF}_{i-1} + (k-1) \times S_i,$$

where $RF_i$ represents the receptive field size of the current layer, and $RF_{i-1}$ denotes the receptive field size of the previous layer. $S_i$ represents the product of the strides of all previous layers (excluding the current layer), and is given by:

$$S_i = \prod_{i=1}^{L_i} stride_i.$$

As shown in Table 4, a larger receptive field in the encoder of neural audio codec brings more contextual information. Although this can enhance audio quality and improve encoding efficiency, it also leads to a significant decline in consistency and gives rise to the DRI phenomenon. Therefore, it is crucial to preserve the original receptive field while allowing the model to balance the trade-offs between audio reconstruction quality and addressing the many-to-one problem.

## 8.2 EVALUATION BASELINES

**SpeechGPT** [2] (Zhang et al., 2023a) is a neural codec language model based on HuBERT (Hsu et al., 2021) with conversational abilities, capable of providing various styles of speech responses based on context and human instructions.

**USLM** [3] (Zhang et al., 2023b) is built upon SpeechTokenizer (Zhang et al., 2023b) and consists of both autoregressive and non-autoregressive models to hierarchically model information in speech. The autoregressive model captures the content information, while the non-autoregressive model complements it by adding paralinguistic information.

**AnyGPT** [4] (Zhan et al., 2024) is an any-to-any multimodal neural codec language model that utilizes discrete representations for various modalities, including speech, text, images, and music. It also uses SpeechTokenizer (Zhang et al., 2023b) to quantize speech.

**VoiceCraft** [5][6] (Peng et al., 2024) is a token-infilling neural codec language model. It introduces a token rearrangement procedure that combines causal masking and delayed stacking to enhance voice cloning ability.

**XTTS v2** [7] (Casanova et al., 2024) is a multilingual speech generation model and employs a VQ-VAE (Van Den Oord et al., 2017) module to discretize the mel-spectrogram.

## 8.3 EVALUATION OF SPEECH GENERATION

To further demonstrate the effectiveness of our method, we conducted validation using different test settings. Following the evaluation testing configuration in VoiceCraft (Peng et al., 2024), we test our method under the same conditions. As shown in Table 5, experimental results show that our method achieve SOTA performance across various testing methods, thereby demonstrating the effectiveness and generality of our method.

## 8.4 CONSISTENCY ACCURACY OF EACH LAYER

As shown in Table 6, we provide a detailed display of the consistency accuracy at each layer for all neural audio codecs, and the accuracy of the neural audio codec with consistency constraint surpasses that of the baseline models at every layer. Specifically, compared to EnCodec (Défossez et al., 2022), our method has shown an average consistency improvement of 21.47%, 29.17%, and 36.29% in the first layer, the first 3 layers, and the first 8 layers, respectively. We can observe that consistency accuracy significantly decreases as the number of layers increases, particularly in baseline models. This may suggest that the semantic information in the shallow layers of codebooks is more relevant to context-independent text, which results in higher consistency accuracy. In contrast, the acoustic information in the deeper layers is more fragile and sensitive, making it more influenced by

---

[2] https://huggingface.co/fnlp/SpeechGPT-7B-com
[3] https://huggingface.co/fnlp/USLM
[4] https://huggingface.co/fnlp/AnyGPT-chat
[5] https://huggingface.co/pyp1/VoiceCraft_giga330M
[6] https://huggingface.co/pyp1/VoiceCraft_830M_TTSEnhanced
[7] https://huggingface.co/coqui/XTTS-v2

Table 5: The speech generation results on LibriTTS test set, following the testing configuration in VoiceCraft. **Bold** means the best result, and underline means the second-best result. **Ours** and **Ours w/o consistency constraint** denote the same neural audio codecs with and without consistency constraint. The subscripts of the neural codec language models (e.g., $330M, 44Kh$) denote the model size and data scale.

| Neural Audio Codec | Bandwidth | Neural Codec Language Model | WER↓ | Objective SIM↑ | UTMOS↑ | Subjective MOS↑ | SMOS↑ |
|---|---|---|---|---|---|---|---|
| Ground Truth | / | / | 0.0 | 71.38% | 4.12 | 4.43 | 4.23 |
| EnCodec | 2.2 kbps | VoiceCraft$_{330M,9Kh}$ | 8.26 | 51.10% | 3.54 | 3.58 | 3.47 |
| | | VoiceCraft$_{830M,9Kh}$ | 4.72 | 55.78% | 3.73 | 3.72 | 3.43 |
| Mel VQ-VAE | / | XTTS_v2$_{27Kh}$ | 3.50 | 60.06% | 3.95 | 3.58 | 3.85 |
| SpeechTokenizer | 4.0 kbps | USLM$_{960h}$ | 8.01 | 56.82% | 3.09 | 3.07 | 2.90 |
| | | AnyGPT$_{57Kh}$ | 25.75 | 25.66% | 3.19 | 2.77 | 2.63 |
| Ours w/o consistency constraint | 4.0 kbps | VALL-E$_{960h}$ | 8.51 | 55.90% | 4.08 | 3.73 | 3.50 |
| | | VALL-E$_{44Kh}$ | 5.11 | 56.20% | 4.12 | 3.92 | 3.40 |
| Ours | 4.0 kbps | VALL-E$_{960h}$ | 3.51 | 60.97% | 4.32 | 3.97 | 3.73 |
| | | VALL-E$_{44Kh}$ | **3.13** | **61.72%** | **4.34** | **4.02** | **3.95** |

Table 6: Detailed results of consistency accuracy of each layer in neural audio codecs. **Ours** denotes the neural audio codec with consistency constraint.

| Neural Audio Codec | Every Layer's Consistency | | | | | | | |
|---|---|---|---|---|---|---|---|---|
| | 1 | 2 | 3 | 4 | 5 | 6 | 7 | 8 |
| EnCodec | 74.66% | 61.20% | 48.62% | 41.30% | 32.47% | 26.30% | 21.25% | 17.89% |
| HiFiCodec | 61.87% | 55.73% | 23.15% | 22.34% | / | / | / | / |
| SpeechTokenizer | 41.52% | 23.13% | 16.09% | 11.64% | 8.59% | 6.21% | 5.08% | 5.31% |
| DAC | 63.44% | 46.17% | 36.88% | 32.77% | 33.75% | 34.92% | 34.26% | 30.90% |
| FunCodec | 29.34% | 10.12% | 7.03% | 4.10% | 2.54% | 1.02% | 0.78% | 0.59% |
| Ours | 96.13% | 91.09% | 84.77% | 79.57% | 73.44% | 68.71% | 63.13% | 57.19% |

context (Zhang et al., 2023b). This could create challenges for downstream neural codec language models when predicting audio tokens from these deeper layers.

## 8.5 EXPLORE GENERALITY OF OUR METHOD ON THE NUMBER OF CODEBOOKS.

Table 7: The speech generation results on LibriTTS test set for VALL-E based on our method with different number of codebooks.

| Neural Audio Codec | | | | Neural Codec Language Model | | | | |
|---|---|---|---|---|---|---|---|---|
| Number of Codebooks | Bandwidth | Slice | Perturbation | WER↓ | Objective SIM↑ | UTMOS↑ | Subjective MOS↑ | SMOS↑ |
| 4 | 2.0 kbps | / | / | 1.93 | 69.86% | 4.29 | 3.88 | 3.35 |
| | | 20% | phase perturb | 1.15 | 72.50% | 4.24 | 4.06 | 3.40 |
| | | 40% | phase perturb | 1.39 | 72.54% | 4.21 | 3.93 | 3.37 |
| 8 | 4.0 kbps | / | / | 4.73 | 76.95% | 4.10 | 3.73 | 3.50 |
| | | 20% | phase perturb | 1.84 | 83.71% | 4.31 | 3.97 | 3.73 |
| | | 40% | phase perturb | 1.90 | 82.81% | 4.27 | 3.84 | 3.70 |

We find that consistency declines significantly with deeper layers. To validate that our method is still effective on shallow layers of codebooks, we evaluate the speech generation performance of VALL-E (Wang et al., 2023) based on our method with different numbers of codebooks. As shown in Table 7, with different numbers of codebooks, the improvement across all evaluation metrics demonstrates the generalizability of our proposed method.

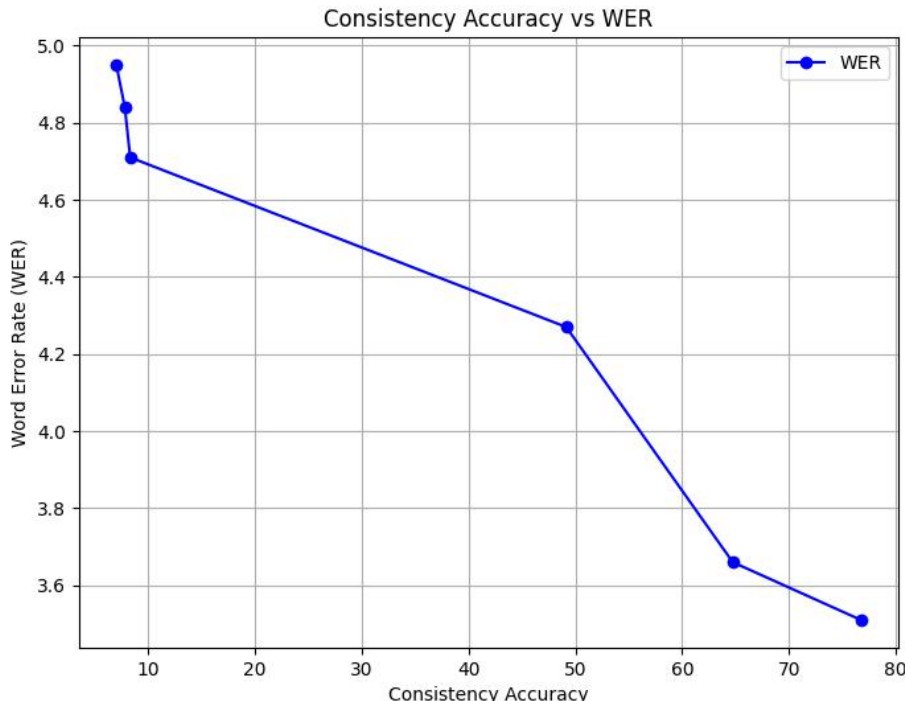

Figure 5: The impact of neural audio codec's consistency accuracy on the downstream VALL-E's WER. The plot demonstrates a clear trend where increasing consistency leads to lower WER.

### 8.6 CORRELATION BETWEEN CONSISTENCY ACCURACY AND WER

As shown in Figure 5, there is a positive correlation between consistency accuracy and WER improvement. Specifically, as the consistency accuracy increases, the WER correspondingly decreases.

### 8.7 MODEL PARAMETERA AND SCALABLE MODEL SIZE

Our neural audio codec utilizes 8 codebooks, each with a vocabulary size of 1024, and a base dimension of 128. It operates at a frame rate of 50Hz for audio sampled at 16kHz, resulting in a total parameter count of 66 M. On average, the speech to be encoded has a duration of 7 seconds, leading to an estimation of FLOPs of approximately $2.57 \times 10^{12}$.

For the neural codec language model, both the AR model and NAR model are built upon the same Transformer architecture. This setup includes 12 layers, 16 attention heads, an embedding dimension of 1024, and a feedforward layer dimension of 4096, collectively comprising about 365M parameters. Typically, in the speech teneration task, the phoneme sequence has a length of 100. The audio tokenizer extracts a sequence of audio tokens with a length of 150 from a 3-second prompt audio, and the generated audio token sequence has a length of 550. This results in a computation of FLOPs approximately equal to $1.66 \times 10^{11}$.

As shown in Table 8, we conducted experiments with models of varying parameter sizes. The observed performance improvements further demonstrate the effectiveness of our method across different model parameter settings.

Table 8: The VALL-E model with different parameter sizes.

| Neural Audio Codec | Total number of parameters | WER↓ | SIM↑ |
|---|---|---|---|
| Ours w/o consistency constraint | 365M | 8.51 | 55.90% |
| Ours with consistency constraint | 365M | 3.50 | 60.97% |
| Ours w/o consistency constraint | 822M | 7.28 | 57.13% |
| Ours with consistency constraint | 822M | 3.16 | 61.33% |

