# OpenReview forum: "Analyzing and Mitigating Inconsistency in Discrete Audio Tokens for Neural Codec Language Models"
_ICLR.cc/2025/Conference — Submitted to ICLR 2025_

### Official Review · Reviewer_XpEs · 2024-10-21

**Soundness:** 3
**Presentation:** 3
**Contribution:** 3
**Rating:** 6
**Confidence:** 3

**Summary:**

The paper addresses the Discrete Representation Inconsistency (DRI) phenomenon in neural audio codecs, where identical audio segments are represented by different discrete token sequences, reducing downstream task performance on speech generation. The authors propose two methods, slice-consistency and perturbation-consistency, to mitigate DRI by aligning token representations from audio slices and perturbed audio. They show significant consistency improvements in popular neural audio codecs and reduced word error rates (WER) in speech generation tasks using large datasets.

**Strengths:**

- The authors identify an interesting problem with audio tokenizers where audio, with or without contextual information,
is encoded by the audio tokenizer into different audio tokens unlike text tokenizer which decode to identical text tokens
- The proposed method aims to retain the original receptive field while enabling the model to address the trade-offs between audio reconstruction quality and resolving the many-to-one problem. To achieve this the method uses a segment of audio randomly sliced, and the encoded representation from this sliced segment is required to closely approximate the corresponding representation obtained from the entire audio. To address many-to-one problem, the representation of an audio and its representation after applying slight spectral
perturbation are forced to be aligned.
- The experiments cover various neural audio codecs and datasets across different bandwidth settings and model sizes, supporting the generality of the proposed solution. The results demonstrate large improvements in consistency metrics (up to 36.29% in deeper layers), WER reduction (3.72%), and speaker similarity increase (5.68%)

**Weaknesses:**

- The paper mainly focuses on consistency-based methods and doesn't extensively compare these approaches to other potential techniques for handling the many-to-one mapping problem, such as the simpler methods introduced in the introduction section (e.g., setting the kernel size to 1). Including such baselines would be helpful for comparison in Table 2.
- The paper would benefit from a baseline without the consistency constraint. The current baselines are different methods/codecs like Encodec, DAC, etc., which might be trained on different datasets. This raises the possibility that the better reconstruction in the author's proposed method might be due to larger training data, rather than the addition of consistency-based loss. To verify this, it would be helpful to include a baseline RVQ-GAN or a model without the additional losses.

**Questions:**

- Can the authors include the two baselines suggested in the weaknesses section? The first is a kernel size of 1, and the second is the baseline performance of a model trained on the same data without consistency loss. If this isn't possible or necessary, could the authors provide a justification?

---

> ### Author Response · Authors · 2024-11-26
> **Response to Reviewer XpEs**
>
> We thank the reviewer for the positive review and constructive feedback, and we hope our response fully resolves your concerns.
>
> ## Question 1:
> The paper mainly focuses on consistency-based methods and doesn't extensively compare these approaches to other potential techniques for handling the many-to-one mapping problem, such as the simpler methods introduced in the introduction section (e.g., setting the kernel size to 1). Including such baselines would be helpful for comparison in Table 2.
>
> **Response 1**: We attempted to set the kernel size of the convolutional layers in the codec to 1. This means that each audio sample would be discretized into an audio token without being influenced by its context, thereby completely avoiding the many-to-one mapping issue. However, this approach introduced other challenges:
>
> (1) Extremely low training and inference efficiency. Typically, a speech segment averages 6 seconds in length and, at a sampling rate of 16,000, consists of 96,000 audio samples. Consequently, the audio token sequence would also be 96,000 tokens long, rendering it impossible to encode such a length of speech in a single pass even on an A100 GPU.
>
> (2) The excessively long audio token sequence is unsuitable for downstream neural codec language models. A sequence of 96,000 audio tokens is exceptionally lengthy, making it difficult for the language model to effectively train and perform inference.
>
> (3) The quality of the reconstructed audio deteriorates, as demonstrated in Table 1 in this section.
>
> Our goal is to reduce inconsistency phenomenon in the codec without compromising its ability to effectively reconstruct audio, ensuring that the audio tokens are more suitable for the language model. Setting the kernel size to 1, while eliminating inconsistency, results in subpar audio reconstruction and generates overly long audio token sequences that are impractical for modeling with neural codec language models.
>
> We appreciate your attention to these points and value your feedback, which is instrumental in refining our approach.
>
> Table 1: neural audio codec trained with different configurations.
>
> | Neural Audio Codec               | ViSQOL | PESQ | Consistency |
> |----------------------------------|--------|------|-------------|
> | Ours with kernel_size = 1        | 3.78   | 2.60 | 98.96%      |
> | Ours with consistency constraint | 4.45   | 3.25 | 71.03%      |
>
>
> ## Question 2
> The paper would benefit from a baseline without the consistency constraint. The current baselines are different methods/codecs like Encodec, DAC, etc., which might be trained on different datasets. This raises the possibility that the better reconstruction in the author's proposed method might be due to larger training data, rather than the addition of consistency-based loss. To verify this, it would be helpful to include a baseline RVQ-GAN or a model without the additional losses.
>
> **Response 2**: Table 1 in our paper may have been misleading. As shown in Table 2 in this section, the total duration of training data for various neural audio codecs is listed. To ensure a fair comparison with other neural audio codecs, we have aligned the bandwidth and training data as closely as possible with theirs. In Table 1 in our paper, the total duration of the training audio for our codec does not exceed that of the other codecs, and in some cases, it is even less. Nevertheless, our codec demonstrates superior reconstruction performance. This highlights that our codec's ability to better reconstruct audio is not due to larger training datasets.
>
> We additionally trained a codec without the consistency loss, and the results are presented in Table 3 in this section. As evidenced by the results, consistency loss does not impact the codec in speech reconstruction. Instead, the consistency loss primarily contributes to enhancing the codec's consistency accuracy.
>
> Table 2: Total duration of training data for various neural audio codecs.
> | Neural Audio Codec | Total Duration |
> |--------------------|----------------|
> | EnCodec            | 2690h          |
> | HiFiCodec          | 1122h          |
> | SpeechTokenizer    | 960h           |
> | DAC                | 2740h          |
> | FunCodec           | 960h           |
> | ours               | 960h           |
>
> Table 3: The speech reconstruction results on LibriTTS test set.
> | Codec                           | ViSQOL | PESQ | Consistency | First 3 Layers’ Consistency |
> |---------------------------------|--------|------|-------------|-----------------------------|
> | Ours w/o consistency constraint | 4.45   | 3.26 | 7.21%       | 17.25%                      |
> | Ours                            | 4.45   | 3.25 | 71.03%      | 88.82%                      |
>
> We greatly value your opinions and hope these revisions adequately address the concerns you previously raised. If you find that our revisions meet your expectations, could you please reassess and update your rating? Thank you for your hard work and assistance.

---

### Official Review · Reviewer_aNo3 · 2024-11-03

**Soundness:** 1
**Presentation:** 2
**Contribution:** 1
**Rating:** 1
**Confidence:** 5

**Summary:**

This paper investigated the problem of Discrete Representation Inconsistency (DRI), where a single audio segment is represented by multiple divergent token sequences. This causes confusion in neural codec language models and leads to omissions and repetitions in speech generation. The authors proposed several consistency regularization techniques that enhance codec reconstruction quality to address this problem. The authors also show that text-to-speech (TTS) systems trained on Encodec using these proposed methods achieve better performance.

**Strengths:**

The problem is interesting.

**Weaknesses:**

1. There is no theoretical evidence presented to support that improved consistency accuracy in codec models leads to better performance in codec and downstream audio LLM tasks. It would be valuable to see further analysis on this aspect. While the authors highlight the consistency improvement achieved by the proposed method, both the evaluation and training methods (i.e., slice method) are consistent. Therefore, the observed improvement is expected.
2. Neural codec language models for music and sound effects generation (e.g., AudioGen, MusicGen) are also highly relevant. Showing the proposed method used for codec and language models on music and sound effect benchmarks and comparing them with state-of-the-art models is essential to establish the effectiveness of the approach.
3. The reported evaluation of TTS models appears inconsistent with existing literature, particularly regarding WER results. It seems likely that re-ranking may have been used before the WER evaluation, where multiple samples are generated, and the one with the best WER is selected. For instance, VALL-E, known for its limitations in generating speech precisely aligned with text transcriptions, typically achieves around 10% WER on LibriSpeech when trained on LibriLight, which can be improved to approximately 5% through re-ranking. In my opinion, the 1.37% WER reported in the paper is a groundbreaking achievement for TTS. While I recognize that the proposed method could enhance TTS robustness, such a substantial improvement is surprising without further modifications to the training pipeline or model architecture. Additionally, VoiceCraft generally achieves around 20% WER on LibriTTS without re-ranking, whereas the results in the paper show a considerably lower WER of about 2-3%. The paper does not specify how TTS samples were generated; if re-ranking was applied, please clarify this and provide results without re-ranking for an accurate comparison.
4. Although the studied question is interesting, its scientific contribution is limited, or at least needs to be supported by more experiments and results.

**Questions:**

1. Please clarify the use of re-ranking methods for TTS evaluation. If re-ranking was used, the authors should report performance across different re-ranking levels (e.g., 5, 10, 20 samples, or no re-ranking) to provide a more comprehensive view of its impact on results.

2. To comprehensively assess the TTS system's robustness, I recommend conducting additional evaluations using challenging sentences, such as "22222222 hello 22222222" (some examples can be found in https://ralle-demo.github.io/RALL-E/ or any reasonable test set is acceptable).

3. The baseline codec models in this work are trained on short utterances (around 1 second). It remains unclear whether training a codec model on longer utterances could improve consistency accuracy. Intuitively, codec models trained on short clips may be more susceptible to acoustic variance (the semantic context to leverage is limited). An additional useful experiment would be to include the Mimi codec (https://github.com/kyutai-labs/moshi) as a baseline, given its training on 12-second speech clips and a 7 million-hour dataset. This comparison could offer valuable insights into the effect of training duration and codec capability on consistency accuracy.

4. I recommend adding more perturbations, such as non-audible noise, reverb, or slight volume augmentation when measuring consistency accuracy. This would help in evaluating the robustness of the model's consistency accuracy under varied conditions.

5.  Please include a breakdown of the WER metrics, specifying insertion, deletion, and replacement errors. As the aim of the proposed method is to address omissions and repetitions during speech generation, a detailed report on these error types is crucial to understanding the system's performance.

6. Further evaluation is recommended on music and sound effects benchmarks to validate the effectiveness of the proposed method, as suggested in the weakness part.

---

> ### Author Response · Authors · 2024-11-26
> **Response to Reviewer aNo3 - part 1**
>
> Thanks for your valuable feedback, and we hope our response fully resolves your concerns.
>
> ## [About Weakness 1]
> **Response 1**: We have responsed this question in the Answer 2 of the Section to all reviewers. In our research, we obtain a series of codecs with various consistency accuracy, which we then utilized as audio tokenizers to train VALL-E and observe how consistency impacts WER. As shown in Table 1 in this section, there is a positive correlation between consistency accuracy and WER improvement. Specifically, as the consistency accuracy increases, the WER correspondingly decreases. To visually illustrate the relationship between WER and consistency, we also plotted the figure<https://github.com/ConsistencyInNeuralCodec/consistencyinneuralcodec.github.io/blob/main/fig/consistency_wer/consistency_wer.jpg>.
>
> As demonstrated in Table 2 in this section, there is a positive correlation between improvements in WER and consistency accuracy of neural audio codecs. For neural audio codecs with the same architecture, we observed a notable reduction in cross-entropy loss when applying consistency constraints during VALL-E training. Moreover, both the top-1 accuracy and top-10 accuracy in predicting the first layer codebook showed enhancements. This suggests that neural audio codecs employing consistency constraints are more suitable for VALL-E.
>
> Table 1: how different consistency accuracy within neural audio codec affect the WER of speech generated by VALL-E.
>
> | Consistency Accuracy | WER  |
> |----------------------|------|
> | 49.15%               | 4.27 |
> | 64.74%               | 3.66 |
> | 76.75%               | 3.51 |
>
> Table 2: Analysis of Consistency Constraints in Neural Audio Codec on VALL-E Performance. This table presents comprehensive metrics including: cross-entropy loss for first codebook layer prediction during training, top-1 accuracy for first audio token prediction, top-10 accuracy (whether the correct audio token appears in top-10 candidates), and inference metrics (WER and SIM) of generated speech.
> | neural audio codec              | consistency accuracy within codecs | cross entropy loss | top1 acc | top10 acc | WER  |
> |---------------------------------|------------------------------------|--------------------|----------|-----------|------|
> | ours w/o consistency constraint | 6.94%                              | 3.384              | 0.247    | 0.651     | 8.51 |
> | ours w consistency constraint   | 64.74%                             | 3.177              | 0.259    | 0.693     | 3.66 |
> | ours w consistency constraint   | 76.75%                             | 3.142              | 0.265    | 0.710     | 3.51 |
>
>
> ## [About Weakness 2]
> **Response 2**: Our study focuses on a novel method utilizing consistency constraints to enhance speech generation by the neural codec language model through the reduction of DRI within the neural audio codec. Our primary objective is specifically targeted at speech generation and does not aim to address challenges related to music and sound effects universally. Our proposed method is designed to be general and applicable across various neural audio codecs. To demonstrate the efficacy of our approach, we conducted validations on the neural codec language model. We emphasize that, theoretically, our method holds applicability across multiple neural codec language models. Additionally, we hope that our findings offer valuable insights and inspiration to other fields.

---

> ### Author Response · Authors · 2024-11-26
> **Response to Reviewer aNo3 - part 2.1**
>
> ## [About Weakness 3 and Question 1]
> **Response 3**: It is considerable that other recent works that have reproduced the results of VALL-E and improved upon them[1, 2, 3, 4, 5], reporting WER significantly below 10%. We have provided the code and training scripts in the links available to all reviewers <https://github.com/ConsistencyInNeuralCodec/ConsistencyInNeuralCodec>. We invite you to utilize these resources to replicate our results and inspect our model. Consequently, we assert that re-ranking was not employed in our methods.
>
> We have responsed this question in the Answer 1 of the Section to all reviewers. In our paper, we compared several TTS systems. And these TTS systems were originally evaluated with various testing configurations in their respective paper [1, 3, 4, 6]. To ensure a strictly fair comparison of these TTS systems, we performed an evaluation on a subset from LibriTTS. We utilized Whisper v3 as the ASR model to measure the WER like XTTS[7], and employed 3D-speaker for SIM evaluation. To further substantiate the effectiveness of our model, we conducted additional experiments following the testing configurations in the VoiceCraft[6]. It is worth noting that VoiceCraft did not utilize re-ranking, achieving a WER of 4.5, which is consistent with our results as shown in Table 3 in this section.
>
> Table 3: The speech generation results on LibriTTS test set. Bold means the best result, and
> underline means the second-best result. **Ours** and **Ours w/o consistency constraint** denote the
> same neural audio codecs with and without consistency constraint. The subscripts of the neural
> codec language models (e.g., 330M, 44Kh) denote the model size and data scale.
>
> | Neural Audio Codec              | Neural Codec Language Model       | WER↓  | SIM↑   | UTMOS↑ |
> |---------------------------------|-----------------------------------|-------|--------|--------|
> | /                               | ground_truth                      | 0.0   | 71.38% | 4.12   |
> | EnCodec                         | VoiceCraft_330m                   | 8.26  | 51.10% | 3.54   |
> |                                 | VoiceCraft_880m                   | 4.72  | 55.78% | 3.73   |
> | Mel VQ-VAE                      | XTTS_v2                           | $\underline{3.50}$  | 60.06% | 3.95   |
> | SpeechTokenizer                 | USLM                              | 8.01  | 56.82% | 3.09   |
> |                                 | AnyGPT                            | 25.75 | 25.66% | 3.19   |
> |                                 | VALL-E (reproduced by Amphion[4]) | 14.32 | 48.90% | 3.87   |
> | ours w/o consistency constraint | VALL-E_960h                       | 8.51  | 55.90% | 4.08   |
> |                                 | VALL-E_44kh                       | 5.11  | 56.20% | 4.12   |
> | ours w consistency constraint   | VALL-E_960h                       | 3.51  | $\underline{60.97%}$ | $\underline{4.32}$   |
> |                                 | VALL-E_44kh                       | **3.13**  | **61.72%** | **4.34**   |
>
>
> [1] Neural Codec Language Models are Zero-Shot Text to Speech Synthesizers
>
> [2] VALL-E 2: Neural Codec Language Models are Human Parity Zero-Shot Text to Speech Synthesizers
>
> [3] VALL-E R: Robust and Efficient Zero-Shot Text-to-Speech Synthesis via Monotonic Alignment
>
> [4] DiTTo-TTS: Efficient and Scalable Zero-Shot Text-to-Speech with Diffusion Transformer
>
> [5] Autoregressive Speech Synthesis without Vector Quantization
>
> [6] VoiceCraft: Zero-Shot Speech Editing and Text-to-Speech in the Wild
>
> [7] XTTS: a Massively Multilingual Zero-Shot Text-to-Speech Model
>
>
>  ## [About Question 4]
>
> **Response 4**: We obtained several hard examples from the VALL-E R demo page <https://www.microsoft.com/en-us/research/project/vall-e-x/vall-e-r/> and tested them on our VALL-E model, as shown in the Table below.

---

> ### Author Response · Authors · 2024-11-26
> **Response to Reviewer aNo3 - part 2.2**
>
> | Text                                                                                                                                                                                                             | Prompt                                                                                                                                              | VALL-E                                                                                                                                                     | VALL-E trained on codec with consistency constraint                                                                                                                    |
> |------------------------------------------------------------------------------------------------------------------------------------------------------------------------------------------------------------------|-----------------------------------------------------------------------------------------------------------------------------------------------------|------------------------------------------------------------------------------------------------------------------------------------------------------------|------------------------------------------------------------------------------------------------------------------------------------------------------------------------|
> | As the cosmic cosmic cosmic cosmic cosmic cosmic dance of the stars unfolds in in in in in in silence, revealing the mystical mysteries of the celestial celestial celestial celestial celestial celestial realm | [1.wav](https://github.com/ConsistencyInNeuralCodec/consistencyinneuralcodec.github.io/blob/main/audio/speech_generation/testset_hard/prompt/1.wav) | [1.wav](https://github.com/ConsistencyInNeuralCodec/consistencyinneuralcodec.github.io/blob/main/audio/speech_generation/testset_hard/result/VALL-E/1.wav) | [1.wav](https://github.com/ConsistencyInNeuralCodec/consistencyinneuralcodec.github.io/blob/main/audio/speech_generation/testset_hard/result/VALL-E-consistency/1.wav) |
> | Beneath the moonlit night, the solitary wolf’s haunting howl howl howl howl howl echoed through the ancient forest, embodying the primal spirit of the wilderness.                                               | [2.wav](https://github.com/ConsistencyInNeuralCodec/consistencyinneuralcodec.github.io/blob/main/audio/speech_generation/testset_hard/prompt/2.wav) | [2.wav](https://github.com/ConsistencyInNeuralCodec/consistencyinneuralcodec.github.io/blob/main/audio/speech_generation/testset_hard/result/VALL-E/2.wav) | [2.wav](https://github.com/ConsistencyInNeuralCodec/consistencyinneuralcodec.github.io/blob/main/audio/speech_generation/testset_hard/result/VALL-E-consistency/2.wav) |
> | As the quantum physicist delved into the quantum realm, the enigmatic entanglement of particles perplexed even the most astute astute astute astute astute astute minds.                                         | [3.wav](https://github.com/ConsistencyInNeuralCodec/consistencyinneuralcodec.github.io/blob/main/audio/speech_generation/testset_hard/prompt/4.wav) | [3.wav](https://github.com/ConsistencyInNeuralCodec/consistencyinneuralcodec.github.io/blob/main/audio/speech_generation/testset_hard/result/VALL-E/4.wav) | [3.wav](https://github.com/ConsistencyInNeuralCodec/consistencyinneuralcodec.github.io/blob/main/audio/speech_generation/testset_hard/result/VALL-E-consistency/4.wav) |
> | Adventurous ants anxiously ate apples, adventurous adventurous apples                                                                                                                                            | [4.wav](https://github.com/ConsistencyInNeuralCodec/consistencyinneuralcodec.github.io/blob/main/audio/speech_generation/testset_hard/prompt/5.wav) | [4.wav](https://github.com/ConsistencyInNeuralCodec/consistencyinneuralcodec.github.io/blob/main/audio/speech_generation/testset_hard/result/VALL-E/5.wav) | [4.wav](https://github.com/ConsistencyInNeuralCodec/consistencyinneuralcodec.github.io/blob/main/audio/speech_generation/testset_hard/result/VALL-E-consistency/5.wav) |

---

> ### Author Response · Authors · 2024-11-26
> **Response to Reviewer aNo3 - part 3**
>
> ## [About Question 5]
> **Response 5**: (1) Our objective in proposing perturbation consistency is to apply slight perturbations to the audio that constrain the consistency of the audio tokens without altering the auditory perception.
>
> (2) There are other perturbation methods, including white noise, background noise, reverberation, and different audio quantization techniques. The essence of these approaches is data augmentation, aimed at enhancing model stability by generating more data. However, these perturbation methods significantly alter the original audio, resulting in a noticeable change in auditory perception. The neural audio codec, being a lightweight model with approximately 66M parameters, struggles with these changes. In early experiments, we also explored other perturbation methods. As shown in Table 4, the quality of reconstruct speech, as well as consistency are decreased.
>
> Table 4: The impact of different perturbation methods on neural audio codecs.
> | Perturb Method | ViSQOL↑ | PESQ↑ | Consistency↑ | First 3 Layers’ Consistency↑ |
> |----------------|---------|-------|--------------|------------------------------|
> | /              | 4.47    | 3.26  | 6.94%        | 15.49%                       |
> | Phase Aug      | 4.46    | 3.26  | 7.03%        | 16.20%                       |
> | White Noise    | 4.26    | 3.17  | 4.37%        | 14.62%                       |
>
>
> ## [About Question 6]
> **Response 6**: As shown in Table 5 in this section, when applying the consistency constraint results, VALL-E generates higher quality speech with a lower WER, with reductions observed in the insertion, substitution, and deletion metrics. Notably, the deletions decrease significantly from 1359 to 405, which might indicate that, without the consistency constraint, VALL-E tends to generate speech with more repeated words.
>
> Table 5: detailed WER information
> | neural codec model              | total WER | insertion | substitution | deletion |
> |---------------------------------|-----------|-----------|--------------|----------|
> | Ours w/o consistency constraint | 8.51      | 213       | 548          | 1359     |
> | Ours w consistency constraint   | 3.50      | 197       | 269          | 405      |

---

> ### Author Response · Authors · 2024-11-28
> **Response to Reviewer aNo3 - part 4**
>
> Initially, because we compare various TTS systems with different testing configurations, we conduct tests on a subset extracted from LibriTTS, utilizing Whisper v3 as the ASR model to evaluate WER[1], and employing the 3D-speaker toolkit to assess SIM[2]. Reviewers mention that our evaluation metrics are not aligned with other papers when presenting our results, which would be better validated under different testing settings as a supplement. Thus, we select a representative work, VoiceCraft[3], and follow its testing configuration to evaluate results on different subsets. Due to the use of different testing settings, including new ASR models (Whisper v2 [3]) and new speaker similarity detection tools(WavLM[4]), it would obviously lead to different results, but our conclusions remain consistent, and the distribution of results for each baseline is also consistent with previous testing settings.
>
> [1] XTTS: a Massively Multilingual Zero-Shot Text-to-Speech Model
>
> [2] 3D-Speaker-Toolkit: An Open Source Toolkit for Multi-modal Speaker Verification and Diarization
>
> [3] VoiceCraft: Zero-Shot Speech Editing and Text-to-Speech in the Wild
>
> [4] WavLM: Large-Scale Self-Supervised Pre-Training for Full Stack Speech Processing

---

> ### Author Response · Authors · 2024-12-02
> **Looking Forward to Your Further Feedback**
>
> Dear Reviewer aNo3,
>
> We sincerely thank you for your valuable suggestions on our work. Based on your insightful feedback, we have provided additional clarifications, and hope our responses have fully addressed your concerns.
>
> We hope you had a wonderful Thanksgiving holiday during the past few days. As the rebuttal deadline approaches, we kindly look forward to receiving your further feedback. Once again, we deeply appreciate the time and effort you have dedicated to reviewing our paper.
>
> Best regards,

---

> ### Comment · Reviewer_aNo3 · 2024-12-02
>
> 1. The newly reported WER of 0% for GT appears suspicious, and the results for VALL-E and VoiceCraft contradict my prior experience. I will refrain from commenting further on this matter.
>
> 2. The authors did not address my third question. Furthermore, the observed correlation between WER and consistency is measured using VALL-E trained with the proposed codec. Several key variables in the codecs, such as training audio lengths, the receptive field of the audio codec, and codec causality, are under-investigated which undermines the clarity and significance of the paper's key contributions.
>
> 3. The authors did not evaluate the proposed method's performance on music audio codecs, despite claiming a focus on speech. However, the paper's title, Discrete Audio Tokens for Neural Codec Language Models, implies a broader scope that extends beyond speech.
>
> 4. The PDF does not appear to have been updated during the rebuttal phase. In its current form, I think the manuscript is not ready for publication.

---

> > ### Author Response · Authors · 2024-12-03
> > **Response to Reviewer aNo3**
> >
> > Thanks for your reply.
> >
> > ## [About Question 2]
> >
> > **Response**:
> >
> > **For your concerns about VALL-E and codec that used to measure consistency:**
> >
> > (1) We train another neural codec language model, UniAudio, with inconsistent or consistent audio tokens. As shown in Table 1 in this section, neural codec language models with consistency constraint demonstrate better performance than those without consistency constraint, which indicates that the consistency constraint is a general method, and is benificial for neural codec language models, including VALL-E, UniAudio, and others, to achieve improvement in intelligibility (WER) and similarity (SIM).
> >
> > Table 1: The speech generation results on LibriTTS test set.
> >
> > | Neural Audio Codec              | Neural Codec Language Model | WER↓ | SIM↑   | UTMOS↑ |
> > |---------------------------------|-----------------------------|------|--------|--------|
> > | ours w/o consistency constraint | UniAudio_960h               | 5.90 | 54.20% | 3.91   |
> > | ours w consistency constraint   | UniAudio_960h               | 2.39 | 59.09% | 4.15   |
> >
> > **For your concerns about key variables in the codecs:**
> >
> > (1) Section 4.1 of the paper details the **training audio lengths** (1.28s) and important training parameters. Ours is non-causal.
> >
> > (2) Section 8.1 in the appendix of the paper lists detailed parameters for **the receptive field of the  codec**, including kernel size, stride, dilation, and receptive field for each layer, as well as the total receptive field.
> >
> > ## [About Question 4]
> >
> > **Response**: Our PDF was updated twice on November 28th. You can check the revision history. Although we can no longer update the PDF at the moment, we will continue to make improvements in the future.

---

> > > ### Comment · Reviewer_aNo3 · 2024-12-03
> > >
> > > Thank you for your reply.
> > >
> > > 1. My concern is not with the LM but rather with the codec. As mentioned, the observed correlation between WER and consistency is based on VALL-E (or UniAudio) trained with the proposed codec. In my opinion, this alone is insufficient to conclude.
> > >
> > > 2. My question pertains to the impact of training audio lengths, the receptive field of the audio codec, and codec causality on consistency accuracy and downstream LM performance - which I believe are important to ablate and analyze.
> > >
> > > 3. In the PDF, some content appears to be outdated, such as the results in Table 2.
> > >
> > > 4. Several of my concerns remain unaddressed.
> > >
> > > Thank you again to the authors for addressing some of my concerns. However, I still do not recommend this paper for acceptance.

---

> > > > ### Author Response · Authors · 2024-12-04
> > > > **Response to Reviewer aNo3 - part 1**
> > > >
> > > > Thanks for your reply.
> > > >
> > > > ## [About Question 1]
> > > >
> > > > **Response**: Besides ours, we conducted additional experiments on EnCodec models. We obtain a series of EnCodec models with different consistency accuracy by varying different weights of consistency constraint loss, which we then utilized as speech tokenizers to train VALL-E on LibriTTS and observe how consistency impacts WER. Table 1 in this section illustrates the effect of various consistency accuracy within EnCodec model on the WER of VALL-E, indicating that there is a positive correlation between consistency accuracy and WER improvement. Specifically, as the consistency accuracy increases, the WER correspondingly decreases.
> > > >
> > > > This reflects the inconsistency phenomenon present in EnCodec models, which can also be mitigated through our proposed method and leads to significant improvements in downstream VALL-E models. This demonstrates that our method is available to applied in different codecs.
> > > >
> > > > Table 1: How different consistency accuracy within EnCodec model affect the WER of speech generated by VALL-E.
> > > >
> > > > | Neural Audio Codec               | Codec's Consistency↑ | Codec's PESQ↑ | VALL-E's WER↓ | VALL-E's SIM↑ |
> > > > |----------------------------------|----------------------|---------------|---------------|---------------|
> > > > | EnCodec                          | 43.51%               | 3.27          | 9.02          | 51.47%        |
> > > > | EnCodec w consistency constraint | 62.98%               | 3.27          | 7.29          | 54.82%        |
> > > > | EnCodec w consistency constraint | 77.29%               | 3.26          | 5.13          | 55.59%        |
> > > >
> > > > ## [About Question 2]
> > > >
> > > > **Response**: (1) In works like VALL-E, UniAudio, and WavTokenizer, they did not explore hyperparameters such as training audio lengths and the receptive field of the audio codec, partly because the impact of these hyperparameters on model improvement is relatively minimal. Therefore, in this paper, we primarily focus on investigating how the Discrete Representation Inconsistency (DRI) phenomenon affects downstream neural codec language models.
> > > >
> > > > (2) We have conducted sufficient experiments on DRI phenomenon, demonstrating both the effectiveness and generalizability of our proposed method.
> > > >
> > > > ## [About Question 3]
> > > >
> > > > **Response**: From the revision history, we can prove that the PDF was updated on November 28th. The results in Table 2 in the paper have been updated in Table 5 in the appendix of the paper. We hold these 2 results for reviewers' and readers' comparison.

---

> > > > ### Author Response · Authors · 2024-12-04
> > > > **Response to Reviewer aNo3 - part 2**
> > > >
> > > > ## [About Question 4]
> > > > **Response:**
> > > >
> > > > I understand your additional concern is about experiments on music and sound effects generation, as you mentioned before.
> > > >
> > > > In this paper, we propose a novel method, consistency constraints, to enhance speech generation by the neural codec language model through the reduction of DRI phenomenon within the neural audio codec. Our primary objective is targeted at speech generation and does not aim to address challenges related to music and sound effects.
> > > >
> > > > Our adequate experiments in the paper and responses have already thoroughly demonstrated that our method can significantly improve the performance of neural codec language models. Does our method have to be validated in the domain of music generation to prove its effectiveness in speech generation? **Is it really necessary that a method proven effective in one domain must also be validated in other domains?** For example, VALL-E[1] only conducted experiments on speech generation tasks, without experiments on music generation tasks. However, many subsequent works[2,3,4] are still based on VALL-E, which doesn't affect VALL-E's usefulness.
> > > >
> > > > [1] Neural Codec Language Models are Zero-Shot Text to Speech Synthesizers
> > > >
> > > > [2] VALL-E 2: Neural Codec Language Models are Human Parity Zero-Shot Text to Speech Synthesizers
> > > >
> > > > [3] VALL-E R: Robust and Efficient Zero-Shot Text-to-Speech Synthesis via Monotonic Alignment
> > > >
> > > > [4] VoiceCraft: Zero-Shot Speech Editing and Text-to-Speech in the Wild

---

### Official Review · Reviewer_eE2Q · 2024-11-03

**Soundness:** 3
**Presentation:** 2
**Contribution:** 3
**Rating:** 5
**Confidence:** 3

**Summary:**

This paper is about analysis and solve the problem of Discrete Representation Inconsistency.

**Strengths:**

1. The paper is well-written and easy to follow.

2. The performance is comparative.

3. The Discrete Representation Inconsistency seems an important question.

**Weaknesses:**

1. Fair Comparison: The authors expanded the training dataset to 44,000 hours. However, in Table 1, I do not think other models, including EnCodec and DAC, were trained on the same data. Is this fair for the results in Table 1?
2. Unprofessional Expressions: The format of the abstracts violates the conference rules. In Figure 3, the word "perception" is presented in both italics and normal text. The equations are not tagged with numbers like "(1), (2)." There is no underline in Table 1, Column "ViSQOL." The bold "Ours and Ours w/o consistency constraint" in the caption of Table 2 should be replaced with quotes. This raises serious questions about the paper, as there is still a long way to go.
3. Efficiency: How large is the model? Is there any insight regarding the parameters and FLOPS? Could the performance boost come from a larger model?
4. Implementation Details: Some important details are not mentioned. The variables related to $\lambda$ in the equation in line 285 are not further detailed in this paper. Are they all set to 1?
5. Novelty: The novelty of this paper is rather weak. There is no structural improvement, only further utilization of the method in Phaseaug. However, the analysis of Discrete Representation Inconsistency is interesting. I consider this as a minor weakness.

**Questions:**

See weaknesses above.

---

> ### Author Response · Authors · 2024-11-26
> **Response to Reviewer eE2Q - part 1**
>
> We are really grateful for your constructive review and valuable feedback, and we hope our response fully resolves your concerns.
>
> ## [About Weakness 1]
>
> **Response1**:  Table 1 in our paper may have been misleading. As shown in Table 3 in this section, the total duration of training data for various neural audio codecs is listed. To ensure a fair comparison with other neural audio codecs, we have aligned the bandwidth and training data as closely as possible with theirs. In Table 1, the total duration of the training audio for our codec does not exceed that of the other codecs, and in some cases, it is even less. Nevertheless, our codec demonstrates superior reconstruction performance. This highlights that our codec's ability to better reconstruct audio is not due to larger training datasets.
>
> Additionally, we have trained VALL-E on 960 hours of audio data. To further validate the effectiveness of our approach on large-scale data, we have extended the training duration for VALL-E to 44,000 hours, while codec was still trained on 960h.
>
> Table 3: Total duration of training data for various neural audio codecs.
> | Neural Audio Codec | Total Duration |
> |--------------------|----------------|
> | EnCodec            | 2690h          |
> | HiFiCodec          | 1122h          |
> | SpeechTokenizer    | 960h           |
> | DAC                | 2740h          |
> | FunCodec           | 960h           |
> | ours               | 960h           |
>
>
> ## [About Weakness 2]
> **Response 2**: Thank you very much for your suggestions on writing and illustration. We will revise the paper according to your recommendations.
>
>
> ## [About Weakness 3]
> **Response 3**: Our codec utilizes 8 codebooks, each with a vocabulary size of 1024, and a base dimension of 128. It operates at a frame rate of 50Hz for audio sampled at 16kHz, resulting in a total parameter count of 66 M. On average, the speech to be encoded has a duration of 7 seconds, leading to an estimation of FLOPs of approximately 2.57 x 10^12.
>
> For the neural codec language model, both the AR model and NAR model are built upon the same Transformer architecture. This setup includes 12 layers, 16 attention heads, an embedding dimension of 1024, and a feedforward layer dimension of 4096, collectively comprising about 365M parameters. Typically, in a TTS task, the phoneme sequence has a length of 100. The audio tokenizer extracts a sequence of audio tokens with a length of 150 from a 3-second prompt audio, and the generated audio token sequence has a length of 550. This results in a computation of FLOPs approximately equal to 1.66 x 10^11.
>
> We conducted experiments with two sets of codecs and VALL-E models differing in parameter size, as shown in Table 4 and Table 5 in this section. As the parameter size increases, both the codec and VALL-E exhibit enhanced performance.
>
> Table 4: codec with different parameter size
>
> | Neural Audio Codec               | Total number of parameter | ViSQOL↑ | PESQ↑ | consistency↑ | First 3 Layers’ Consistency↑ |
> |----------------------------------|---------------------------|---------|-------|--------------|------------------------------|
> | ours w/o consistency constraint  | 66M                       | 4.47    | 3.25  | 6.94%        | 15.49%                       |
> | ours with consistency constraint | 66M                       | 4.45    | 3.25  | 71.03%       | 88.82%                       |
> | ours w/o consistency constraint  | 110M                      | 4.67    | 3.39  | 6.82%        | 16.31%                       |
>
> Table 5: VALL-E with different parameter size
>
> | Neural Audio Codec               | Total number of parameter | WER  | SIM    |
> |----------------------------------|---------------------------|------|--------|
> | Ours w/o consistency constraint  | 365M                      | 8.51 | 55.90% |
> | Ours with consistency constraint | 365M                      | 3.50 | 60.97% |
> | Ours w/o consistency constraint  | 822M                      | 7.28 | 57.13% |
> | Ours with consistency constraint | 822M                      | 3.16 | 61.33% |
>
>
> ## [About Weakness 4]
> **Response 4**:  In the loss function described in Section 4.2, the parameters are set as $ \lambda_{adv} = 0.11$, $ \lambda_{fm} = 11.11$, and $ \lambda_{rvq} = 1.0$. When applying consistency constraints to the codec, $ \lambda_{con} = 10.0$.

---

> ### Author Response · Authors · 2024-11-26
> **Response to Reviewer eE2Q - part 2**
>
> ## [About Weakness 5]
> **Response 5**: This paper is a research-oriented work, where we focuse on a novel method utilizing consistency constraints to enhance speech generation by the neural codec language model through the reduction of DRI phenomenon within the neural audio codec. Our primary objective is specifically targeted at the analysis and the mitigation of DRI phenomenon, and does not aim to improve the model structure. Our proposed method is designed to be general and applicable across various neural audio codecs. To demonstrate the efficacy of our approach, we conducted validations on the neural codec language model. Additionally, we hope that our findings offer valuable insights and inspiration to other fields.
>
> Thank you for your thorough review and valuable suggestions on our paper. Based on your feedback, we have made the corresponding revisions and improvements.
>
> We greatly value your opinions and hope that these revisions adequately address the issues and concerns you previously raised. If you find that our revisions meet your expectations, could you please reassess and update your rating? Once again, thank you for your hard work and assistance.

---

> > ### Comment · Reviewer_eE2Q · 2024-11-27
> > **Thanks for your response**
> >
> > Thank you for your diligent experiments and thorough responses. While your answers address my questions, there are still outstanding concerns.
> >
> > The clarification of questions Q1, Q2, Q3, and Q4 should facilitate straightforward revisions to the paper. However, I noticed that the manuscript has not been updated for two weeks of discussion, and there is even a blank space to be used for clarification on the 10th page.
> >
> > After reading other reviews, it appears there is work to be done before the paper is ready for publication. Therefore, my score will remain 5.

---

> ### Author Response · Authors · 2024-11-28
> **Response to Reviewer eE2Q**
>
> We sincerely appreciate your valuable feedback and suggestions regarding the writing and formatting of the paper. We have addressed the identified issues and have uploaded a revised version of the PDF. Our major revisions are outlined as follows:
>
> (1) We have added the training duration for each neural audio codec in Table 1 in the paper of the original paper using subscripts.
>
> (2) We have corrected the format of the abstract, the numbers of the mathematical equations and other formatting issues.
>
> (3) In Section 8.7 in the appendix, we report the total number of parametes and FLOPs for the neural audio codec and VALL-E, respectively. Additionally, we examine how VALL-E's performance varies with different parameter sizes.
>
> (4) In Section 4.1 in the paper, we have included the specific values of the weight $\lambda$ used in the loss function.
>
> We still hope you can avoid being influenced by certain reviewers. Our code has been uploaded to <https://github.com/ConsistencyInNeuralCodec/ConsistencyInNeuralCodec>, and you are welcome to reproduce and verify our results.
>
> Once again, thank you for your diligent efforts in reviewing our work.

---

> ### Author Response · Authors · 2024-12-02
> **Looking Forward to Your Further Feedback**
>
> Dear Reviewer eE2Q,
>
> We sincerely thank you for your valuable suggestions on our work. Based on your insightful feedback, we have provided additional clarifications, and hope our responses have fully addressed your concerns.
>
> We hope you had a wonderful Thanksgiving holiday during the past few days. As the rebuttal deadline approaches, we kindly look forward to receiving your further feedback. Once again, we deeply appreciate the time and effort you have dedicated to reviewing our paper.
>
> Best regards,

---

> ### Author Response · Authors · 2024-12-04
> **Response to Reviewer eE2Q**
>
> We would like to express our gratitude again for your time and effort in reviewing our paper. If you find that our revisions meet your expectations or address some of your concerns, could you please reassess and update your rating?

---

### Official Review · Reviewer_2NT9 · 2024-11-04

**Soundness:** 2
**Presentation:** 2
**Contribution:** 2
**Rating:** 6
**Confidence:** 3

**Summary:**

This paper  introduces and analyzes the Discrete Representation Inconsistency (DRI) phenomenon, which occurs in discrete audio tokens. DRI leads to confusion in large language models (LLMs) during prediction, causing omissions and repetitions in speech generation. The inconsistency arises because the same audio segment can be encoded into different discrete sequences depending on the context, complicating the model’s prediction process.

To address this issue, the authors propose two solutions:

Slice-Consistency Method: This method ensures that a randomly sliced segment of audio has an encoded representation consistent with that of the entire audio. It helps maintain consistency without reducing the model's ability to understand the audio.

Perturbation-Consistency Method: This method aligns the encoded representations of audio before and after slight spectral perturbations, enhancing the model's robustness. The perturbations are subtle enough to be imperceptible to human hearing but help the model remain stable against small variations.

The authors conduct extensive experiments to validate the effectiveness of these methods. The results demonstrate that these approaches significantly alleviate the DRI phenomenon and improve speech generation performance. Metrics such as reduced Word Error Rate (WER), increased speaker similarity, and enhanced naturalness of generated speech confirm the success of the proposed solutions.

**Strengths:**

1. The paper introduces a new problem, Discrete Representation Inconsistency (DRI), in discrete audio tokens for neural codec language models, proposing two effective methods, Slice-Consistency and Perturbation-Consistency, to improve token consistency while maintaining audio quality.

2. Thorough experiments demonstrate that the proposed methods effectively reduce DRI and improve key metrics like Word Error Rate (WER) and speaker similarity, validated on both small (LibriTTS) and large (MLS) datasets.

3. The paper is well-organized and easy to follow, clearly explaining the DRI problem, proposed solutions, and their effectiveness, making the study accessible and understandable.

**Weaknesses:**

1. Lack of Direct Evidence Linking Codec Inconsistency to Performance：The paper says that codec inconsistency causes omissions and repetitions in outputs, but there’s no direct evidence showing how much this affects performance. While the DRI issue is clear, its exact impact on results isn’t. Adding experiments that compare models trained with consistent and inconsistent audio tokens would make this point stronger.

2. Alternative Perturbations and the Importance of Perturbation-Consistency：The paper doesn’t talk about whether other types of perturbations, besides spectral perturbations, could help improve consistency. It also doesn’t clearly explain why Perturbation-Consistency is necessary or how it works. Trying out other types of perturbations and explaining why spectral perturbations were chosen would make the method more convincing.

**Questions:**

1. The paper assumes that inconsistencies between sliced and full audio representations in codecs lead directly to issues like omissions and repetitions in LLM outputs. However, there is a lack of detailed evidence or analysis showing the extent of this impact. While codec inconsistencies are highlighted, it’s unclear how strongly this correlates with LLM performance issues. Text tokenizers, which don’t face consistency problems, still experience omissions and repetitions, suggesting that other factors could be influencing these issues. More evidence linking codec inconsistency to LLM performance would strengthen this argument.

2. The results in Table 3 show that the scenario with no slicing but with phase perturbation has the lowest consistency (around 7%), yet achieves the second-best WER. This outcome contradicts the paper’s central argument that higher consistency improves model performance. This suggests that low consistency doesn’t necessarily result in poor performance, which weakens the foundation of the paper’s claims.

3. The performance difference between SpeechTokenizer and EnCodec in Table 2 is unusually large, diverging significantly from the results in the original SpeechTokenizer paper[1]. This raises concerns about the fairness and consistency of the comparisons. Without clear justification for these differences, the validity of the conclusions drawn from these comparisons is questionable. The setup for these evaluations needs more transparency to ensure fair and reliable comparisons.
[1] Zhang X, Zhang D, Li S, et al. "Speechtokenizer: Unified speech tokenizer for speech large language models," *arXiv preprint*, arXiv:2308.16692, 2023.

---

> ### Author Response · Authors · 2024-11-26
> **Response to Reviewer 2NT9 - part1**
>
> We thank the reviewer for the constructive and professional review.
>
> ## [About Weakness 1 and Question 1]
>
> **Reponse 1**: We demonstrate the impact of the codec with consistency constraint and without consistency constraint on downstream VALL-E, as shown in Table 1 in this section, indicating that speech generated by the codec with consistency constraint have reductions in insertion, substitution, and deletion.
>
> To more intuitively illustrate the DRI issue, we trained codec with different configurations of slice perturbation (slice percentages) to obtain codecs with different consistency, and utilized these codecs as audio tokenizers and trained VALL-E to observe how consistency affects the WER, as shown in Table 2 in this section and the figure <https://github.com/ConsistencyInNeuralCodec/consistencyinneuralcodec.github.io/blob/main/fig/consistency_wer/consistency_wer.jpg>, in the **Answer 2** to the **Section to all reviewers**. Specifically, as the consistency accuracy increases, the WER correspondingly decreases.
>
> Table 2 in our paper demonstrates the pipelines of ours (with or without consistency constraint) + VALL-E, which presents models trained with inconsistent and consistent audio tokens. Compared to the VALL-E model trained on inconsistent audio tokens, VALL-E trained on consistent audio tokens achieves significant improvement in WER and similarity. This indicates that consistent audio tokens leads to better TTS performance.
>
> Table 1: detailed WER information.
> | neural codec model              | total WER | insertion | substitution | deletion |
> |---------------------------------|-----------|-----------|--------------|----------|
> | Ours w/o consistency constraint | 8.51      | 213       | 548          | 1359     |
> | Ours w consistency constraint   | 3.50      | 197       | 269          | 405      |
>
>
> Table 2: how different consistency accuracy within neural audio codec affect the WER of speech generated by VALL-E.
> | Consistency Accuracy | 49.15 | 64.74 | 76.75% |
> | Consistency Accuracy | WER  |
> |----------------------|------|
> | 49.15%               | 4.27 |
> | 64.74%               | 3.66 |
> | 76.75%               | 3.51 |
>
>
> ## [About Weakness 2]
>
> **Reponse 2**: Our goal is to enforce audio token consistency while preserving perceptual audio quality. Phase perturbation achieves this by introducing minimal modifications that are imperceptible to human listeners.
>
> Also, we explored various perturbation techniques, including white noise, reverberation and different audio quantization methods. While these traditional data augmentation approaches aim to improve model robustness, our early experiments (shown in Table 3 in this section) revealed that they significantly degrade audio reconstruction quality and reduce consistency metrics and ViSQOL scores.  Data augmentation approaches are particularly challenging for our lightweight neural audio codec (approximately 70M parameters).
>
> In our practice, we found that phase perturbation demonstrates significant advantages over alternative perturbation approaches. The relationship between spectral phase and temporal information is fundamental: phase information determines the temporal arrangement of different frequency components in a signal, thereby affecting the temporal waveform. However, phase modifications have minimal impact on speech perceptual characteristics.
>
> This unique property creates a valuable opportunity: the audio details affected by phase perturbation are largely irrelevant for both human perception and downstream VALL-E modeling. As a result, we can leverage phase perturbation for data augmentation without compromising the codec's reconstruction capabilities.
>
> Table 3: The impact of different perturbation methods on the neural audio codec.
>
> | Perturb Method | ViSQOL↑ | PESQ↑ | Consistency↑ | First 3 Layers’ Consistency↑ |
> |----------------|---------|-------|--------------|------------------------------|
> | /              | 4.47    | 3.26  | 6.94%        | 15.49%                       |
> | Phase Aug      | 4.46    | 3.26  | 7.03%        | 16.20%                       |
> | White Noise    | 4.26    | 3.17  | 4.37%        | 14.62%                       |
>
> | Perturb Method | ViSQOL↑ | PESQ↑ | Consistency↑ | First 3 Layers’ Consistency↑ |
> |----------------|---------|-------|--------------|------------------------------|
> | /              | 4.47    | 3.26  | 6.94%        | 15.49%                       |
> | Phase Aug      | 4.46    | 3.26  | 7.03%        | 16.20%                       |
> | White Noise    | 4.26    | 3.17  | 4.37%        | 14.62%                       |

---

> ### Author Response · Authors · 2024-11-26
> **Response to Reviewer 2NT9 - part2**
>
> ## [About Question 2]
> **Reponse 3**:
>
> We have also observed this scenario and have conducted additional experiments to further investigate it. By varying the weight of perturbation consistency, we obtained codecs with various consistency accuracies, allowing us to study the relationship between consistency and WER when only perturbation consistency is considered. Our findings indicate that, compared to slice consistency, perturbation consistency has a smaller impact on consistency accuracy. However, as shown in Table 4, improving consistency through phase perturbation still enhances the performance of downstream VALL-E in generating audio. For a clearer illustration, we have also included the figure<https://github.com/ConsistencyInNeuralCodec/consistencyinneuralcodec.github.io/blob/main/fig/consistency_wer/consistency_wer_phase_aug.jpg>.
>
> We speculate that phase perturbation may partially decouple information within the audio. Specifically, the phase perturbation affect the temporal arrangement of audio signals, altering the structure of some information. This may effectively prevent certain features from overfitting to details not prominent for the model, thereby improving WER performance. At the same time, this suggests that certain phase information of audio signals is not decisive for the neural codec language model's prediction of audio tokens. Therefore, even with lower consistency in some experimental settings, a good WER can still be achieved.
>
> This result highlights that, while increasing consistency generally improves performance of generated speech, we also need to consider the complexity of other audio features, such as phase information, and their potential impact on the neural codec language model. We will update the paper to include these discussions and appreciate you for pointing out this observation, which helps improve our research.
>
> Table 4: how different consistency accuracy under different phase perturbation within neural audio codec affect the WER of speech generated by VALL-E.
>
> | Consistency Accuracy | WER  |
> |----------------------|------|
> | 7.03%                | 4.95 |
> | 7.82%                | 4.84 |
> | 8.33%                | 4.71 |

---

> ### Author Response · Authors · 2024-11-26
> **Response to Reviewer 2NT9 - part3**
>
> ## [About Question 3]
> **Reponse 4**: In Table 1 in our paper, pipelines of SpeechTokenizer + USLM, SpeechTokenizer + AnyGPT are based on open-source models, and the pipeline of SpeechTokenizer + VALL-E is reproduced by Amphion[1].
> The pipelines of ours (with or without consistency constraint) + VALL-E are independently reproduced by us. We strictly adhered to the original VALL-E[1] paper's model architecture and training parameters. Our reproduced VALL-E demonstrated better speech generation metrics compared to the original VALL-E, not inferior results. Therefore, when assessing the consistency constraints, our experiments conducted on our own VALL-E reproduction provide stronger evidence. Additionally, we have included the experimental results of Amphion's[1] reproduced VALL-E for your reference and comparison.
>
> In our study, we compared various TTS systems. And the original papers presenting these TTS systems employed different testing configurations [2, 3, 4]. To provide a fair and unbiased comparison, we conducted tests on a subset extracted from LibriTTS, utilizing the Whisper v3 as the ASR model to evaluate WER like XTTS[4],  and employing the 3D-speaker toolkit to assess SIM. Additionally, we have followed the testing configuration outlined in VoiceCraft[2] for further experimentation to validate our model's effectiveness. The results of these experiments are detailed in Table 1 in the Answer 2 to the Section to all reviewers.
>
> Table 5: The speech generation results on LibriTTS test set. Bold means the best result, and
> underline means the second-best result. **Ours** and **Ours w/o consistency constraint** denote the
> same neural audio codecs with and without consistency constraint. The subscripts of the neural
> codec language models (e.g., 330M, 44Kh) denote the model size and data scale.
>
> | Neural Audio Codec              | Neural Codec Language Model       | WER↓  | SIM↑   | UTMOS↑ |
> |---------------------------------|-----------------------------------|-------|--------|--------|
> | /                               | ground_truth                      | 0.0   | 71.38% | 4.12   |
> | EnCodec                         | VoiceCraft_330m                   | 8.26  | 51.10% | 3.54   |
> |                                 | VoiceCraft_880m                   | 4.72  | 55.78% | 3.73   |
> | Mel VQ-VAE                      | XTTS_v2                           | $\underline{3.50}$  | 60.06% | 3.95   |
> | SpeechTokenizer                 | USLM                              | 8.01  | 56.82% | 3.09   |
> |                                 | AnyGPT                            | 25.75 | 25.66% | 3.19   |
> |                                 | VALL-E (reproduced by Amphion[4]) | 14.32 | 48.90% | 3.87   |
> | ours w/o consistency constraint | VALL-E_960h                       | 8.51  | 55.90% | 4.08   |
> |                                 | VALL-E_44kh                       | 5.11  | 56.20% | 4.12   |
> | ours w consistency constraint   | VALL-E_960h                       | 3.51  | $\underline{60.97%}$ | $\underline{4.32}$   |
> |                                 | VALL-E_44kh                       | **3.13**  | **61.72%** | **4.34**   |
>
> [1] Amphion: An Open-Source Audio, Music, and Speech Generation Toolkit
>
> [2] Neural Codec Language Models are Zero-Shot Text to Speech Synthesizers
>
> [3] VoiceCraft: Zero-Shot Speech Editing and Text-to-Speech in the Wild
>
> [4] XTTS: a Massively Multilingual Zero-Shot Text-to-Speech Model
>
> Thank you for your thorough review and valuable suggestions on our paper. We greatly value your opinions and hope that these revisions adequately address the issues and concerns you previously raised. If you find that our revisions meet your expectations, could you please reassess and update your rating? Once again, thank you for your hard work and assistance.

---

> ### Author Response · Authors · 2024-12-02
> **Looking Forward to Your Further Feedback**
>
> Dear Reviewer 2NT9,
>
> We sincerely thank you for your valuable suggestions on our work. Based on your insightful feedback, we have provided additional clarifications, and hope our responses have fully addressed your concerns.
>
> We hope you had a wonderful Thanksgiving holiday during the past few days. As the rebuttal deadline approaches, we kindly look forward to receiving your further feedback. Once again, we deeply appreciate the time and effort you have dedicated to reviewing our paper.
>
> Best regards,

---

> ### Author Response · Authors · 2024-12-04
> **Response to Reviewer 2NT9**
>
> We would like to express our gratitude again for your time and effort in reviewing our paper. If you find that our revisions meet your expectations or address some of your concerns, could you please reassess and update your rating?

---

### Official Review · Reviewer_iYgA · 2024-11-05

**Soundness:** 3
**Presentation:** 3
**Contribution:** 2
**Rating:** 5
**Confidence:** 5

**Summary:**

The paper investigates the phenomenon of Discrete Representation Inconsistency (DRI) in audio token sequences generated by neural audio codecs. Unlike text tokens, which are deterministic, discrete audio tokens can vary significantly even if the perceptual audio remains identical. This inconsistency complicates the prediction of subsequent tokens in neural codec language models, leading to potential errors in audio generation. The authors propose two novel methods to address DRI: the slice-consistency and perturbation-consistency methods. They demonstrate the effectiveness of these methods through extensive experiments, showing significant improvements in token consistency, reduced Word Error Rate (WER), and enhanced speaker similarity in speech synthesis tasks.

**Strengths:**

1. The proposed methods are easy to understand and straightforward to implement, which enhances their practical applicability.

2. The identification and in-depth analysis of Discrete Representation Inconsistency in neural audio codecs are novel contributions that highlight an important challenge in the field.

3. The methods substantially enhance the consistency of audio tokens, leading to better performance in speech synthesis tasks, as evidenced by significant improvements in WER and speaker similarity metrics.

**Weaknesses:**

1, Missing Baseline Comparison

In your discussion of WER improvements, your paper lacks a comprehensive comparison with other works that have also improved VALL-E to reduce WER. Since many follow-up studies on VALL-E aim to lower WER, it’s important to systematically compare your method with these works to highlight your approach's advantages.

For example, MELLE [1] uses mel-spectrogram to replace codec enhancing model robustness. VALL-E 2 [2] proposes the repetition aware sampling and grouped code modeling to enhance the stability.


2, Evaluation Metric

Your paper uses different WER and SIM metrics compared to those used in the original VALL-E (WavLM-TDNN,  HuBERT-Large-finetune) and its subsequent improvement papers. This inconsistency makes it difficult for readers to understand the extent of your improvements, leading to confusion. It’s recommended to align your metrics with those in the original VALL-E paper for clearer comparative analysis.

3, Fairness of Baseline Experiments


There are concerns about the fairness of your baseline experiments. It is unclear whether the training configurations for your VALL-E model using your codec were the same as those for the baseline VALL-E models, such as the one using SpeechTokenizer. Ensuring consistent training settings is crucial for a fair comparison.

Additionally, it is puzzling that the WER of your baseline codec outperforms that of SpeechTokenizer. From my understanding, VALL-E with SpeechTokenizer typically achieves a lower WER compared to standard codecs.


[1] Meng L, Zhou L, Liu S, et al. Autoregressive speech synthesis without vector quantization[J]. arXiv preprint arXiv:2407.08551, 2024.


[2] Chen S, Liu S, Zhou L, et al. VALL-E 2: Neural Codec Language Models are Human Parity Zero-Shot Text to Speech Synthesizers[J]. arXiv preprint arXiv:2406.05370, 2024.

**Questions:**

Consistency Accuracy and WER Correlation

Is consistency accuracy truly correlated with WER? In Figure 4, SpeechTokenizer performs worse than EnCodec and DAC in terms of consistency accuracy, which seems counterintuitive. From my experience, when training VALL-E, SpeechTokenizer typically yields better WER results compared to EnCodec and DAC.

---

> ### Author Response · Authors · 2024-11-26
> **Response to Reviewer iYgA - part 1**
>
> We sincerely thank you for your thorough examination of our paper and your valuable feedback.
>
>
> ## [About Weakness 1]
>
> **Response 1**: Our research proposes a method to enhance neural audio codecs, theoretically applicable to various commonly used neural audio codecs. By reducing inconsistencies in the encoding process, we can improve the performance of downstream neural codec language models, such as VALL-E[1] and VALL-E 2[2], rather than being limited to VALL-E alone.
>
> (1) The improved versions of VALL-E[1], such as VALL-E 2[2] and VALL-E R[3], primarily focus on model architecture enhancements. In contrast, our approach targets the improvement of neural audio codecs and is a more general solution that can be broadly applied to various neural codec language models. Our method is orthogonal to the improvements made in VALL-E, meaning that when used together with the improved versions of VALL-E[1, 2, 3], it can bring further advancements in speech generation.
>
> (2) Compared to VALL-E's strategy of modeling based on discrete audio tokens, MELLE[4] employs an auto-regressive approach to model continuous representations and generate Mel-spectrograms. MELLE[4] is fundamentally different from our methodology, as we focus on addressing the inconsistency issues within neural audio codecs.
>
> [1] Neural Codec Language Models are Zero-Shot Text to Speech Synthesizers
>
> [2] VALL-E 2: Neural Codec Language Models are Human Parity Zero-Shot Text to Speech Synthesizers
>
> [3] VALL-E R: Robust and Efficient Zero-Shot Text-to-Speech Synthesis via Monotonic Alignment
>
> [4] Autoregressive Speech Synthesis without Vector Quantization
>
>
> ## [About Weakness 2]
>
> **Reponse 2**: In our study, we compared various TTS systems. And the original papers presenting these TTS systems employed different testing configurations [1, 2, 3]. To provide a fair and unbiased comparison, we conducted tests on a subset extracted from LibriTTS, utilizing the Whisper v3 as the ASR model to evaluate WER like XTTS[3], and employing the 3D-speaker toolkit to assess SIM. Additionally, we have followed the testing configuration outlined in VoiceCraft[2] for further experimentation to validate our model's effectiveness. The results of these experiments are detailed in Table 1 in this section.
>
> Furthermore, to ensure the reproducibility of our results, we have made our code and training scripts publicly available here <https://github.com/ConsistencyInNeuralCodec/ConsistencyInNeuralCodec>.
>
> Table 1: The speech generation results on LibriTTS test set. Bold means the best result, and
> underline means the second-best result. **Ours** and **Ours w/o consistency constraint** denote the
> same neural audio codecs with and without consistency constraint. The subscripts of the neural
> codec language models (e.g., 330M, 44Kh) denote the model size and data scale.
>
> | Neural Audio Codec              | Neural Codec Language Model       | WER↓  | SIM↑   | UTMOS↑ |
> |---------------------------------|-----------------------------------|-------|--------|--------|
> | /                               | ground_truth                      | 0.0   | 71.38% | 4.12   |
> | EnCodec                         | VoiceCraft_330m                   | 8.26  | 51.10% | 3.54   |
> |                                 | VoiceCraft_880m                   | 4.72  | 55.78% | 3.73   |
> | Mel VQ-VAE                      | XTTS_v2                           | $\underline{3.50}$  | 60.06% | 3.95   |
> | SpeechTokenizer                 | USLM                              | 8.01  | 56.82% | 3.09   |
> |                                 | AnyGPT                            | 25.75 | 25.66% | 3.19   |
> |                                 | VALL-E (reproduced by Amphion[4]) | 14.32 | 48.90% | 3.87   |
> | ours w/o consistency constraint | VALL-E_960h                       | 8.51  | 55.90% | 4.08   |
> |                                 | VALL-E_44kh                       | 5.11  | 56.20% | 4.12   |
> | ours w consistency constraint   | VALL-E_960h                       | 3.51  | $\underline{60.97%}$ | $\underline{4.32}$   |
> |                                 | VALL-E_44kh                       | **3.13**  | **61.72%** | **4.34**   |
>
> [1] Neural Codec Language Models are Zero-Shot Text to Speech Synthesizers
>
> [2] VoiceCraft: Zero-Shot Speech Editing and Text-to-Speech in the Wild
>
> [3] XTTS: a Massively Multilingual Zero-Shot Text-to-Speech Model
>
> [4] Amphion: An Open-Source Audio, Music, and Speech Generation Toolkit

---

> ### Author Response · Authors · 2024-11-26
> **Response to Reviewer iYgA - part 2**
>
> ## [About Weakness 3]
>
> **Response 3**: In Table 1 of our paper, pipelines of SpeechTokenizer + USLM, SpeechTokenizer + AnyGPT are based on open-source models, and the pipeline of SpeechTokenizer + VALL-E is reproduced by Amphion[1].  The pipelines of ours (with or without consistency constraint) + VALL-E are independently reproduced by us. We strictly adhered to the model configurations outlined in the VALL-E paper[1]. Specifically, both the AR model and NAR model are built upon the same Transformer architecture. This setup includes 12 layers, 16 attention heads, an embedding dimension of 1024, and a feedforward layer dimension of 4096. Our reproduced VALL-E demonstrated better speech generation metrics compared to the original VALL-E, not inferior results. Therefore, when assessing the consistency constraints, our experiments conducted on our own VALL-E reproduction provide stronger evidence. Additionally, we have included the experimental results of Amphion's[1] reproduced VALL-E for your reference and comparison.
>
> [1] Amphion: An Open-Source Audio, Music, and Speech Generation Toolkit
>
> [2] Neural Codec Language Models are Zero-Shot Text to Speech Synthesizers
>
>
> ## Question 1: Consistency Accuracy and WER Correlation
> **Reponse 4**: In our research, we conducted experiments with altering the consistency perturbation during the training process of the codec, and resulted in a series of codecs with various consistency accuracy, which we then utilized as audio tokenizers to train VALL-E and observe how consistency impacts WER.
>
> As demonstrated in Table 3 in this section, there is a positive correlation between improvements in WER and consistency accuracy of neural audio codecs. For neural audio codecs with the same architecture, we observed a notable reduction in cross-entropy loss when applying consistency constraints during VALL-E training. Moreover, both the top-1 accuracy and top-10 accuracy in predicting the first layer codebook showed enhancements. This suggests that neural audio codecs employing consistency constraints are more suitable for VALL-E.
>
> Table 4 in this section illustrates the effect of various consistency accuracy within neural audio codecs on the WER of VALL-E. As shown Table 4 in this section, there is a positive correlation between consistency accuracy and WER improvement. Specifically, as the consistency accuracy increases, the WER correspondingly decreases. To visually illustrate the relationship between WER and consistency, we also plotted the figure <https://github.com/ConsistencyInNeuralCodec/consistencyinneuralcodec.github.io/blob/main/fig/consistency_wer/consistency_wer.jpg>.
>
> Table 3: Analysis of Consistency Constraints in Neural Audio Codec on VALL-E Performance. This table presents comprehensive metrics including: cross-entropy loss for first codebook layer prediction during training, top-1 accuracy for first audio token prediction, top-10 accuracy (whether the correct audio token appears in top-10 candidates), and inference metrics (WER and SIM) of generated speech.
>
> | neural audio codec              | consistency accuracy within codecs↑ | cross entropy loss↓ | top1 acc↑ | top10 acc↑ | WER↓ |
> |---------------------------------|-------------------------------------|---------------------|-----------|------------|------|
> | ours w/o consistency constraint | 6.94%                               | 3.384               | 0.247     | 0.651      | 8.51 |
> | ours w consistency constraint   | 64.74%                              | 3.177               | 0.259     | 0.693      | 3.66 |
> | ours w consistency constraint   | 76.75%                              | 3.142               | 0.265     | 0.710      | 3.51 |
>
> Table 4: how different consistency accuracy within neural audio codec affect the WER of speech generated by VALL-E.
>
> | Consistency Accuracy | WER  |
> |----------------------|------|
> | 49.15%               | 4.27 |
> | 64.74%               | 3.66 |
> | 76.75%               | 3.51 |
>
>
> ## Question 5:
>
> In Figure 4, SpeechTokenizer performs worse than EnCodec and DAC in terms of consistency accuracy, which seems counterintuitive. From my experience, when training VALL-E, SpeechTokenizer typically yields better WER results compared to EnCodec and DAC.
>
> **Reponse 5**: While our research emphasizes the importance of consistency, we acknowledge that other codec characteristics also play crucial roles in the WER of speech generated by neural codec language models. A notable example is SpeechTokenizer, where the first layer of audio tokens primarily encodes semantic information with minimal acoustic content. This semantic-focused encoding enable VALL-E to effectively model the first layer tokens, achieving better performance  of generated speech, despite lower consistency accuracy.

---

> > ### Comment · Reviewer_iYgA · 2024-11-27
> >
> > Thank you for your detailed responses to my review.  However, I still have some concerns.
> >
> > 1. You mention that your method is theoretically applicable to various neural audio codecs and is not limited to VALL-E alone. However, it appears that your experiments are conducted solely with VALL-E.
> >
> > 2. The issue regarding evaluation metrics remains unresolved, and other reviewers have also raised this concern.
> >
> > 3. You compared your results with those reproduced by Amphion, which is not an official reproduction of VALL-E. I'm still uncertain about the validity of using these results for comparison.
> >
> > 4. The correlation between consistency accuracy and WER seems to be evident only in your own experiments and does not appear to hold for other codecs.

---

> ### Author Response · Authors · 2024-11-28
> **Response to Reviewer iYgA - part1**
>
> Thank you so much for your valuable comments and precious time.
>
> ## [About Question 1]
> **Response**: We are conducting this experiment, and we will report experimental results before the deadline.
>
> ## [About Question 2]
> **Response**: Initially, because we compare various TTS systems with different testing configurations, we conduct tests on a subset extracted from LibriTTS, utilizing Whisper v3 as the ASR model to evaluate WER[1], and employing the 3D-speaker toolkit to assess SIM[2]. Reviewers mention that our evaluation metrics are not aligned with other papers when presenting our results, which would be better validated under different testing settings as a supplement. Thus, we select a representative work, VoiceCraft[3], and follow its testing configuration to evaluate results on different subsets. Due to the use of different testing settings, including new ASR models (Whisper v2 [3]) and new speaker similarity detection tools(WavLM[4]), it would obviously lead to different results, but our conclusions remain consistent, and the distribution of results for each baseline is also consistent with previous testing settings.
>
> [1] XTTS: a Massively Multilingual Zero-Shot Text-to-Speech Model
>
> [2] 3D-Speaker-Toolkit: An Open Source Toolkit for Multi-modal Speaker Verification and Diarization
>
> [3] VoiceCraft: Zero-Shot Speech Editing and Text-to-Speech in the Wild
>
> [4] WavLM: Large-Scale Self-Supervised Pre-Training for Full Stack Speech Processing
>
> ## [About Question 3]
> **Response**:  In the Section to All Reviewers, there are two VALL-E reproductions from different works: SpeechTokenizer+VALL-E reproduced by Amphion[1], and Ours+VALL-E, which is our reproduction. Our reproduced VALL-E demonstrated better speech generation metrics compared to the original VALL-E, not inferior results. Therefore, when assessing the consistency constraints, our experiments conduct on our own VALL-E reproduction provide stronger evidence. And we still report the result of SpeechTokenizer+VALL-E in hope to provide reviewers and readers with a reference.
>
> Although we have provided a detailed explanation of the origins of these two VALL-E models in the last version of the paper's PDF,  maybe you are still confused by this comparison between two reproduced VALL-E models. Considering that your suggestion makes sense, as comparing the open-source VALL-E with other models could cause unnecessary misunderstandings, and since the open-source VALL-E does not affect our experimental conclusions, we have removed the open-source VALL-E from the TTS systems to be compared in the new version of the PDF.
>
> [1] Amphion: An Open-Source Audio, Music, and Speech Generation Toolkit

---

> ### Author Response · Authors · 2024-12-02
> **Response to Reviewer iYgA - part2**
>
> ## [About Question 1]
>
> **Response**: In order to reply your concerns, we conducted additional experiments.
>
> We train another neural codec language model, UniAudio, with inconsistent or consistent audio tokens. As shown in Table 1 in this section, neural codec language models with consistency constraint demonstrate better performance than those without consistency constraint, which indicates that the consistency constraint is a general method, and is benificial for neural codec language models, including VALL-E, UniAudio, and others, to achieve improvement in intelligibility (WER) and similarity (SIM).
>
> We hope our experimental results will be helpful for your assessment.
>
> Table 1: The speech generation results on LibriTTS test set.
>
> | Neural Audio Codec              | Neural Codec Language Model | WER↓ | SIM↑   | UTMOS↑ |
> |---------------------------------|-----------------------------|------|--------|--------|
> | ours w/o consistency constraint | UniAudio_960h               | 5.90 | 54.20% | 3.91   |
> | ours w consistency constraint   | UniAudio_960h               | 2.39 | 59.09% | 4.15   |
>
>
> ## [About Question 4]
>
> **Reponse:** Besides ours, we conducted additional experiments on EnCodec models. We obtain a series of EnCodec models with different consistency accuracy by varying different weights of consistency constraint loss, which we then utilized as audio tokenizers to train VALL-E and observe how consistency impacts WER. Table 1 in this section illustrates the effect of various consistency accuracy within EnCodec model on the WER of VALL-E, indicating that there is a positive correlation between consistency accuracy and WER improvement. Specifically, as the consistency accuracy increases, the WER correspondingly decreases.
>
> This reflects the inconsistency phenomenon present in EnCodec models, which can also be mitigated through our proposed method and leads to significant improvements in downstream VALL-E models. This demonstrates that our method is available to applied in different codecs.
>
> Table 1: How different consistency accuracy within EnCodec model affect the WER of speech generated by VALL-E.
>
> | Neural Audio Codec               | Codec's Consistency↑ | Codec's PESQ↑ | VALL-E's WER↓ | VALL-E's SIM↑ |
> |----------------------------------|----------------------|---------------|---------------|---------------|
> | EnCodec                          | 43.51%               | 3.27          | 9.02          | 51.47%        |
> | EnCodec w consistency constraint | 62.98%               | 3.27          | 7.29          | 54.82%        |
> | EnCodec w consistency constraint | 77.29%               | 3.26          | 5.13          | 55.59%        |

---

> ### Author Response · Authors · 2024-12-02
> **Looking Forward to Your Further Feedback**
>
> Dear Reviewer iYgA,
>
> We sincerely thank you for your valuable suggestions on our work. Based on your insightful feedback, we have provided additional clarifications, and hope our responses have fully addressed your concerns.
>
> We hope you had a wonderful Thanksgiving holiday during the past few days. As the rebuttal deadline approaches, we kindly look forward to receiving your further feedback. Once again, we deeply appreciate the time and effort you have dedicated to reviewing our paper.
>
> Best regards,

---

> ### Author Response · Authors · 2024-12-04
> **Response to Reviewer iYgA**
>
> We would like to express our gratitude again for your time and effort in reviewing our paper. If you find that our revisions meet your expectations or address some of your concerns, could you please reassess and update your rating?

---

### Author Response · Authors · 2024-11-26
**To all reviewers - part1**

Thanks for your valuable feedback, and we hope our response fully resolves your concerns.

## Question 1:
For **Reviewer iYgA**'s Weakness 2: Evaluation Metric
Your paper uses different WER and SIM metrics compared to those used in the original VALL-E (WavLM-TDNN, HuBERT-Large-finetune) and its subsequent improvement papers. This inconsistency makes it difficult for readers to understand the extent of your improvements, leading to confusion. It’s recommended to align your metrics with those in the original VALL-E paper for clearer comparative analysis.

For **Reviewer 2NT9**'s Question 3:
The performance difference between SpeechTokenizer and EnCodec in Table 2 is unusually large, diverging significantly from the results in the original SpeechTokenizer paper[1]. This raises concerns about the fairness and consistency of the comparisons. Without clear justification for these differences, the validity of the conclusions drawn from these comparisons is questionable. The setup for these evaluations needs more transparency to ensure fair and reliable comparisons.

__Reponse 1:__ In our study, we compared various TTS systems. And the original papers presenting these TTS systems employed different testing configurations [1, 2, 3]. To provide a fair and unbiased comparison, we conducted tests on a subset extracted from LibriTTS, utilizing the Whisper v3 as the ASR model to evaluate WER like XTTS[3], and employing the 3D-speaker toolkit to assess SIM. Additionally, to validate our model's effectiveness, we have followed the testing configuration outlined in VoiceCraft[2] for further experimentation. The results of these experiments are detailed in Table 1 in this section.

Furthermore, to ensure the reproducibility of our results, we have made our code and training scripts publicly available here <https://github.com/ConsistencyInNeuralCodec/ConsistencyInNeuralCodec>.

Table 1: The speech generation results on LibriTTS test set. Bold means the best result, and
underline means the second-best result. **Ours** and **Ours w/o consistency constraint** denote the
same neural audio codecs with and without consistency constraint. The subscripts of the neural
codec language models (e.g., 330M, 44Kh) denote the model size and data scale.

| Neural Audio Codec              | Neural Codec Language Model       | WER↓  | SIM↑   | UTMOS↑ |
|---------------------------------|-----------------------------------|-------|--------|--------|
| /                               | ground_truth                      | 0.0   | 71.38% | 4.12   |
| EnCodec                         | VoiceCraft_330m                   | 8.26  | 51.10% | 3.54   |
|                                 | VoiceCraft_880m                   | 4.72  | 55.78% | 3.73   |
| Mel VQ-VAE                      | XTTS_v2                           | $\underline{3.50}$  | 60.06% | 3.95   |
| SpeechTokenizer                 | USLM                              | 8.01  | 56.82% | 3.09   |
|                                 | AnyGPT                            | 25.75 | 25.66% | 3.19   |
|                                 | VALL-E (reproduced by Amphion[4]) | 14.32 | 48.90% | 3.87   |
| ours w/o consistency constraint | VALL-E_960h                       | 8.51  | 55.90% | 4.08   |
|                                 | VALL-E_44kh                       | 5.11  | 56.20% | 4.12   |
| ours w consistency constraint   | VALL-E_960h                       | 3.51  | $\underline{60.97%}$ | $\underline{4.32}$   |
|                                 | VALL-E_44kh                       | **3.13**  | **61.72%** | **4.34**   |

[1] Neural Codec Language Models are Zero-Shot Text to Speech Synthesizers

[2] VoiceCraft: Zero-Shot Speech Editing and Text-to-Speech in the Wild

[3] XTTS: a Massively Multilingual Zero-Shot Text-to-Speech Model

[4] Amphion: An Open-Source Audio, Music, and Speech Generation Toolkit

---

> ### Comment · Reviewer_aNo3 · 2024-11-26
>
> Thank you for your response.
>
> I observed that the TTS results posted here do not align with those reported in the submission paper, and the current results seem to be worse. As the authors claimed they did not use re-ranking, could you please explain the gap?
>
> For example:
>
> | Model                              | Metric | Old WER (%) | New WER (%) |
> |------------------------------------|--------|-------------|-------------|
> | VoiceCraft 330M 9kHz               | WER    | 2.57        | 8.26        |
> | VoiceCraft 330M 9kHz               | WER    | 2.80        | 4.72        |
> | Proposed (w. consistency) VALL-E   | WER    | 1.84        | 3.51        |
> | Proposed (w. consistency) VALL-E   | WER    | 1.37        | 3.13        |
> | Ground Truth (GT)                  | WER    | 1.37        | 0.00        |

---

> ### Author Response · Authors · 2024-11-28
> **Response to Reviewer aNo3**
>
> Initially, because we compare various TTS systems with different testing configurations, we conduct tests on a subset extracted from LibriTTS, utilizing Whisper v3 as the ASR model to evaluate WER[1], and employing the 3D-speaker toolkit to assess SIM[2]. Reviewers mention that our evaluation metrics are not aligned with other papers when presenting our results, which would be better validated under different testing settings as a supplement. Thus, we select a representative work, VoiceCraft[3], and follow its testing configuration to evaluate results on different subsets. Due to the use of different testing settings, including new ASR models (Whisper v2 [3]) and new speaker similarity detection tools(WavLM[4]), it would obviously lead to different results, but our conclusions remain consistent, and the distribution of results for each baseline is also consistent with previous testing settings.
>
> [1] XTTS: a Massively Multilingual Zero-Shot Text-to-Speech Model
>
> [2] 3D-Speaker-Toolkit: An Open Source Toolkit for Multi-modal Speaker Verification and Diarization
>
> [3] VoiceCraft: Zero-Shot Speech Editing and Text-to-Speech in the Wild
>
> [4] WavLM: Large-Scale Self-Supervised Pre-Training for Full Stack Speech Processing

---

### Author Response · Authors · 2024-11-26
**To all reviewers - part2**

Thanks for your valuable feedback, and we hope our response fully resolves your concerns.

## Question 2:

For **Reviewer iYgA**'s question 1: Consistency Accuracy and WER Correlation

For **Reviewer 2NT9**'s Weakness 1: Lack of Direct Evidence Linking Codec Inconsistency to Performance
The paper says that codec inconsistency causes omissions and repetitions in outputs, but there’s no direct evidence showing how much this affects performance. While the DRI issue is clear, its exact impact on results isn’t. Adding experiments that compare models trained with consistent and inconsistent audio tokens would make this point stronger.

For **Reviewer aNo3**'s weakness 1: There is no theoretical evidence presented to support that improved consistency accuracy in codec models leads to better performance in codec and downstream audio LLM tasks. It would be valuable to see further analysis on this aspect. While the authors highlight the consistency improvement achieved by the proposed method, both the evaluation and training methods (i.e., slice method) are consistent. Therefore, the observed improvement is expected.

__Reponse 2:__ In our research, we obtain a series of codecs with various consistency accuracy, which we then utilized as audio tokenizers to train VALL-E and observe how consistency impacts WER. As shown in Table 2 in this section, there is a positive correlation between consistency accuracy and WER improvement. Specifically, as the consistency accuracy increases, the WER correspondingly decreases. To visually illustrate the relationship between WER and consistency, we also plotted the figure <https://github.com/ConsistencyInNeuralCodec/consistencyinneuralcodec.github.io/blob/main/fig/consistency_wer/consistency_wer.jpg>.

Table 2: how different consistency accuracy within neural audio codec affect the WER of speech generated by VALL-E.
| Consistency Accuracy | 49.15 | 64.74 | 76.75% |
| Consistency Accuracy | WER  |
|----------------------|------|
| 49.15%               | 4.27 |
| 64.74%               | 3.66 |
| 76.75%               | 3.51 |

---

### Author Response · Authors · 2024-11-28
**To all reviewers - part3**

In the part 1 of Section to All Reviewers, there are two VALL-E reproductions from different works: SpeechTokenizer+VALL-E reproduced by Amphion[1], and Ours+VALL-E, which is our reproduction. Our reproduced VALL-E demonstrated better speech generation metrics compared to the original VALL-E, not inferior results. Therefore, when assessing the consistency constraints, our experiments conduct on our own VALL-E reproduction provide stronger evidence. And we still report the result of SpeechTokenizer+VALL-E in hope to provide reviewers and readers with a reference.

Although we have provided a detailed explanation of the origins of these two VALL-E models in the last version of the paper's PDF, Reviewer iYgA is still confused by this comparison between two reproduced VALL-E models. Considering that Reviewer iYgA's suggestion makes sense, as comparing the open-source VALL-E with other models could cause unnecessary misunderstandings, and since the open-source VALL-E does not affect our experimental conclusions, we have removed the open-source VALL-E from the TTS systems to be compared in the new version of the PDF.

[1] Amphion: An Open-Source Audio, Music, and Speech Generation Toolkit

---

### Meta-Review · Area_Chair_xoCx · 2024-12-22

**Metareview:**

> The paper investigates the phenomenon of Discrete Representation Inconsistency (DRI) in audio token sequences generated by neural audio codecs. Unlike text tokens, which are deterministic, discrete audio tokens can vary significantly even if the perceptual audio remains identical. This inconsistency complicates the prediction of subsequent tokens in neural codec language models, leading to potential errors in audio generation. The authors propose two novel methods to address DRI: the slice-consistency and perturbation-consistency methods. They demonstrate the effectiveness of these methods through extensive experiments, showing significant improvements in token consistency, reduced Word Error Rate (WER), and enhanced speaker similarity in speech synthesis tasks.

Reviewer aNo3 is unduly harsh as the authors addressed most of his concerns. However, even by totally discounting reviewer aNo3 score, the paper has scores 6-5-5 and should be rejected. Several of the weaknesses raised by reviewers are real. Even though the paper improved significantly (with more experiments) during the rebuttal period, it is more likely the paper could have been accepted if in its current state from submission time. The initial version was lacking several experiments and explanations.

**Additional Comments On Reviewer Discussion:**

Only Reviewers aNo3 and iYgA took part in the rebuttal period. Neither changed their mind.

---

### Decision · Program_Chairs · 2025-01-22

Reject